# UMAP: A Highly Extensible and Physics-Based Simulation Environment for Multi-Agent Reinforcement Learning

## Abstract

Existing simulation environments in the field of multi-agent reinforcement learning (MARL) either lack authenticity or complexity. The data generated by these environments significantly deviate from the requirements of the real world, hindering the practical application of MARL. To address this issue, we propose *Unreal Multi-Agent Playground* (UMAP), a highly extensible, physics-based 3D simulation environment implemented on the Unreal Engine. UMAP is user-friendly in terms of deployment, modification, and visualization, and all its components are open-sourced[1]. Based on UMAP, we design a series of MARL tasks featuring heterogeneous agents, large-scale agents, multiple teams, and sparse team rewards. We also develop an experimental framework compatible with algorithms ranging from rule-based to MARL-based provided by third-party frameworks. In the experimental section, we utilize the designed tasks to test several state-of-the-art algorithms. Additionally, We also conduct a physical experiment to demonstrate UMAP's potential in sim-to-real applications, which is a significant advantage due to the high extensibility and authenticity of UMAP. We believe UMAP can play an important role in the MARL field by evaluating existing algorithms and helping them apply to real-world scenarios, thus advancing the field of MARL.

## 1 Introduction

Multi-agent reinforcement learning (MARL) has demonstrated remarkable potential in many practical fields, including swarm robotic control (Kalashnikov et al., 2018; Chen et al., 2020), autonomous vehicles (Peng et al., 2021b), and video games (Vinyals et al., 2019; Chen et al., 2022). However, a peculiar phenomenon can be observed in the field of MARL (Oroojlooy & Hajinezhad, 2023): although numerous new algorithms are claimed to achieve state-of-the-art (SOTA) performance every year, algorithms actually utilized in real-world applications tend to be classic MARL algorithms or extensions of single-agent reinforcement learning (SARL) algorithms, such as IQL (Tan, 1993) and IPPO (Schulman et al., 2017). Some studies even find that the performance of SARL algorithms in certain multi-agent scenarios outperforms that of some MARL algorithms (Papoudakis et al.). This indicates that the development of MARL has encountered a bottleneck, with many algorithms performing well only in specific simulated tasks but struggling to be applied in real-world scenarios.

One of the keys to breaking through this bottleneck lies in the data of MARL. As a data-driven approach, MARL depends on high-quality data for the design and evaluation of its algorithms. However, if the data distribution is far from that of real-world problems, current developments fail to align with practical needs. As a learning approach driven by rewards, the data of MARL originates from various simulation environments, and that's where the problem lies.

The existing simulation environments of MARL are either overly simplistic and lack authenticity, or limited to low-complexity decision-making, thereby failing to fully reflect the unique challenges of MARL. For instance, MAgent (Zheng et al., 2018) and GoBigger (Zhang et al., 2022) have the capability to support large-scale multi-agent and multi-team training respectively, but the state transitions in these environments are simply achieved through interaction rules among particle-like

---

[1]During the review phase, we put the main codes in the supplementary material, and details of open source statement can be found in Appendix A.

agents, which prevents them from fully simulating real-world conditions. Other examples include the widely used environments like Starcraft Multi-Agent Challenge (SMAC) (Samvelyan et al., 2019) and Google Research Football (GRF) (Kurach et al., 2020), which can simulate scenarios of video game and soccer respectively, offering a certain level of authenticity but with decision-making complexity far below real-world requirements (Zhang et al., 2022). To address this issue, it is unrealistic to develop an all-inclusive environment, but developing an environment which is both extensible and authentic holds significant value.

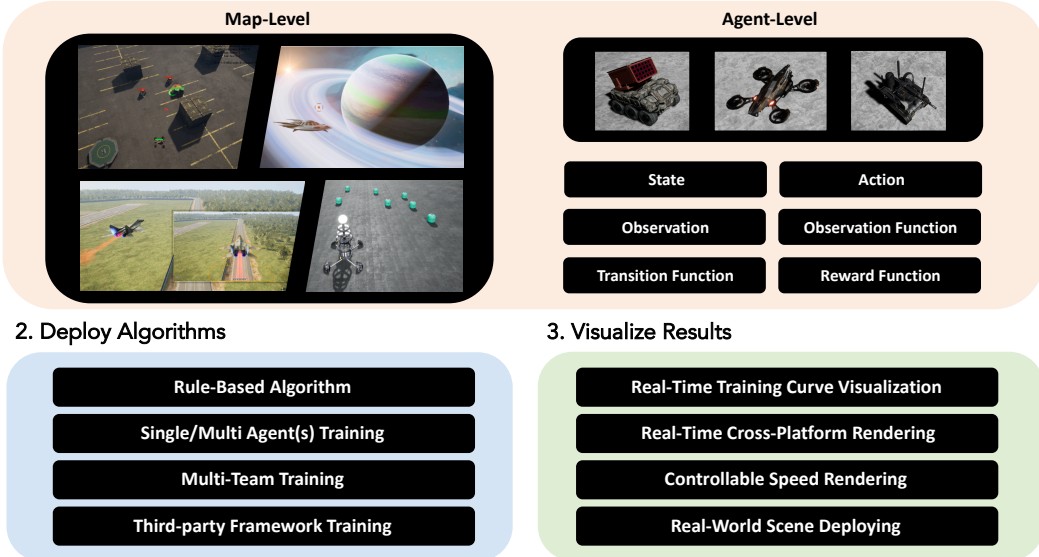

Figure 1: The research workflow for using UMAP. For novice users, UMAP provides direct access to built-in maps and tasks, and offers comprehensive result visualization capabilities. For advanced users, UMAP enables the modification of built-in tasks or the creation of new tasks to test research ideas, and even the deployment of trained algorithms in real-world settings.

In this paper, we propose *Unreal Multi-Agent Playground* (UMAP) to fill this gap. UMAP is a highly extensible, physics-based 3D simulation environment implemented on the Unreal Engine (UE). Compared to existing commonly used environments, UMAP offers four primary advantages: (1) **Support for diversified multi-agent tasks**, UMAP includes a variety of built-in tasks such as heterogeneous-agent tasks, large-scale multi-agent tasks, and multi-team tasks, providing users with a broad selection of tasks to choose from. (2) **Customizable multi-agent task design**, UMAP provides interfaces that allow users to conveniently customize all task properties, such as observations, actions, and state transitions. (3) **Controllable simulation time flow**, users can control the simulation speeds, enabling them to accelerate simulations to expedite training or decelerate simulations for slow-motion analysis. (4) **Rich rendering mechanisms**, UMAP supports controllable-speed rendering and cross-platform real-time rendering (e.g., training on Linux and rendering on Windows simultaneously). The detailed comparison of UMAP and other related works can be found in Table 1 and Appendix B.

To fully utilize the capabilities of UMAP, we also develop an MARL experimental framework known as the Hybrid Multi-Agent Playground (HMAP). This framework includes implementations of rule-based algorithms, built-in MARL algorithms, and algorithms from third-party frameworks such as PyMARL2 (Hu et al., 2021) and HARL (Zhong et al., 2024). By leveraging UMAP and HMAP, users can rapidly customize and deploy environments and algorithms, validate new research ideas, and even apply them in practical scenarios. The overview of the research workflow for using UMAP is depicted in Figure 1.

Our contributions can be summarized as four main parts: firstly, a fully open-source and highly extensible UE-based MARL environment; secondly, an accompanying modular MARL experimental framework; thirdly, a collection of typical multi-agent tasks (covering heterogeneous, large-scale, multi-team, sparse team rewards tasks, and a sim-to-real demo); fourthly, pre-deployed basic algo-

rithms along with experimental analysis based on the above tasks. We believe UMAP can serve as a comprehensive tool to advance the development of MARL and ultimately facilitate their application in real-world scenarios.

Table 1: Comparison of UMAP with other related MARL simulation environments.

| | UMAP(Ours) | MPE | MAgent | Hanabi | NeuralMMO | GoBigger | JaxMARL |
|---|---|---|---|---|---|---|---|
| Heterogenous Support | ✓ | ✓ | ✓ | – | ✓ | – | ✓ |
| Large-Scale Support[2] | ✓ | – | ✓ | – | ✓ | ✓ | ✓ |
| Multi-Team Support | ✓ | – | – | – | – | ✓ | – |
| Mixed-Game Support[3] | ✓ | ✓ | ✓ | – | ✓ | ✓ | ✓ |
| 3D Physics Engine | ✓ | – | – | – | – | – | – |
| Fully Open Source[4] | ✓ | ✓ | ✓ | ✓ | – | – | ✓ |
| All Elements Customizable[5] | ✓ | ✓ | ✓ | ✓ | – | – | ✓ |
| Controllable Time-flow Speed | ✓ | – | – | – | – | – | – |
| Rendering Training[6] | ✓ | – | – | – | ✓ | – | – |

| | GRF | SMAC | SMACv2 | Hide-and-Seek | HoK3v3 | MAMuJoCo | Marathon |
|---|---|---|---|---|---|---|---|
| Heterogenous Support | ✓ | ✓ | ✓ | – | ✓ | ✓ | ✓ |
| Large-Scale Support | – | – | – | – | – | ✓ | – |
| Multi-Team Support | – | – | – | – | – | – | – |
| Mixed-Game Support | – | ✓ | ✓ | – | – | – | – |
| 3D Physics Engine | ✓ | ✓ | ✓ | ✓ | ✓ | ✓ | ✓ |
| Fully Open Source | ✓ | – | – | ✓ | – | ✓ | – |
| All Elements Customizable | – | – | – | – | – | – | – |
| Controllable Time-flow Speed | – | – | – | – | – | – | – |
| Rendering Training | – | – | – | – | – | – | ✓ |

## 2 BACKGROUND

To accommodate various interaction relationships among multi-agent and multi-team scenarios (Fu et al., 2024), we use Partially Observable Markov Game (POMG) (Littman, 1994; Gronauer & Diepold, 2022) to model the MARL problem. A POMG can be represented by an 8-tuple $\langle N, \{S^i\}_{i \in N}, \{O^i\}_{i \in N}, \{\Omega^i\}_{i \in N}, \{A^i\}_{i \in N}, \{\mathcal{T}^i\}_{i \in N}, r, \gamma \rangle$. $N$ is the set of all agents, $\{S^i\}_{i \in N}$ is the global state space which can be factored as $\{S^i\}_{i \in N} = \times_{i \in N} S^{(i)} \times S^E$, where $S^{(i)}$ is the state space of an agent $i$, and $S^E$ is the environmental state space, corresponding to all the non-agent entities. $\{O^i\}_{i \in N} = \times_{i \in N} O^{(i)}$ is the joint observation space and $\{\Omega^i\}_{i \in N}$ is the set of observation functions. Similarly, $\{A^i\}_{i \in N}$ is the joint action space of all agents. $\{\mathcal{T}^i\}_{i \in N}$ is the collection of all agents' transitions and the environmental transition. Finally, $\gamma$ is the discount factor and $r : \{S^i\}_{i \in N} \times \{A^i\}_{i \in N} \times N \to \mathbb{R}$ is the agent-level reward function.

We define *team* as a collection of agents, which all share the same overall goal in a purely cooperative form. Agents within the same team aim to find an optimal joint policy that maximizes the cumulative reward for the whole team. Denoting the joint policy of a certain team $A \subseteq N$ as $\bar{\pi}_A$, the optimal policy $\bar{\pi}_A^*$ can be represented as:

$$\bar{\pi}_A^* = \arg\max_{\bar{\pi}_A} \mathbb{E}_{\bar{\pi}_A} \left[ \sum_{k=0}^{\infty} \gamma^k \sum_{i \in A} r_{t+k}^i \mid \bar{s}_t = \bar{s} \right], \qquad (1)$$

where $\bar{s}$ is the initial global state, $\gamma^k \sum_{i \in A} r_{t+k}^i$ is the discounted return of team $A$, $r_{t+k}^i$ is the reward of an agent $i \in A$ at timestep $t + k$.

---

[2]In this paper, we refer to scenarios involving more than 100 agents (excluding non-agent entities) as large-scale scenarios.

[3]Mixed-game support refers to the simulation environment's capability to support competitive, cooperative, and mixed interaction relationships among agents in scenarios.

[4]All simulation components are open source. Using SMAC as a counterexample, its back-end Starcraft II, cannot be accessed or modified by researchers.

[5]All elements of the POMG (see details in Section 2) related to the environment can be modified.

[6]Remotely connecting to non-render client running inside a server via network, and rendering the on-going training process locally via TCP&UDP.

# 3 UMAP

## 3.1 BASIC CONCEPTS IN UMAP

Multi-agent simulation can demonstrate great diversity in different domains. In order to break the limitations of existing environments, it is necessary to introduce a few new concepts that align with human intuition as well as the requirements of multi-agent simulation.

**Agents** and **Teams:** Agents are the basic decision-making units in the environments. UMAP introduces a new concept "team" to distinguish agents with different goals. UMAP supports numbers of teams, where teams may engage in competition or cooperation. Each team possesses its own independent goal and is equipped with a separate learning-based (or rule-based) algorithm.

**Tasks** and **Scenarios:** Tasks corresponds to POMGs defined in Section 2. The properties of tasks in UMAP include the types and numbers of agents, their team affiliations, as well as each agent's state, observations, reward functions, etc. A scenario can give rise to a series of tasks, which typically share similar reward functions, implying that the objectives to be achieved by the multi-agent systems are the same.

**Maps:** Maps in UMAP determine where the task takes place. A map can be *a small room, or a city full of buildings*. It is a great advantage that UMAP decouples the concept of tasks and maps, as users can conveniently deploy a task in new maps (as long as the agent has the appropriate size and a suitable position initialization function).

**Entities:** Entities are objects in simulation that do not make decisions but still has important functionality. For instance *a street lamp or a dynamic obstacle*. A shared characteristic of these objects is that they must be removed or reinitialized when an episode ends or a new episode starts.

**Events:** We define an event system to simplify the reward crafting procedure. For instance, an event will be generated when an agent is destroyed or an episode is ended. When it is time to compute next-step reward, these events will provide convenient reference.

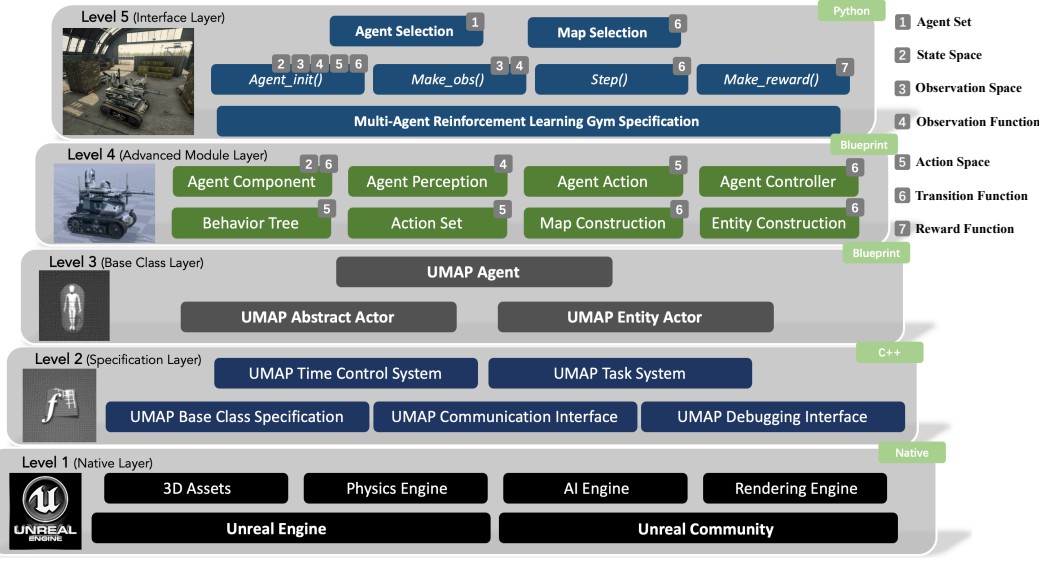

Figure 2: Architecture of UMAP. UMAP employs a hierarchical, five-layered architecture, all of which are open source. Users can modify all elements within POMG by configuring parameters through the Python-based *interface layer*. For more advanced development requirements, users can conveniently adjust scenario elements using graphical programming through the *advanced module layer*.

## 3.2 Utilizing UMAP to customize tasks

UMAP employs a hierarchical five-layer architecture, where each layer builds upon the previous one. From bottom to top, the five layers are: *native layer*, *specification layer*, *base class layer*, *advanced module layer*, and *interface layer*. **Users only need to focus on the *advanced module layer* and the *interface layer*.** In most cases, modifying the basic functions and configuration parameters in the *interface layer* is sufficient to alter all elements of tasks.

Figure 2 shows the internal structure of each layer of UMAP and the task elements affected by each submodule. Specifically, the *native layer* includes 3D assets from the Unreal community and the Unreal Engine, some part of which have been optimized for MARL compatibility. The *specification layer* consists of UMAP's underlying systems and programming specifications, all implemented in C++. The *base class layer* includes all basic classes implemented using Blueprints[7]. These three layers form the foundation of UMAP.

The *advanced module layer*, also based on Blueprints, allows for the modification of agents' physical properties such as appearances, perceptions and kinematics, thereby enabling the development of various agents. This layer also facilitates the development of environmental entities and maps. The top layer is the *interface layer*, implemented in Python and compliant with the gym standard. It includes basic functions like *reset*, *step*, and *done*. Additionally, it supports customizable observations and reward functions. This layer also allows for the selection of maps and agents. More details about the UMAP architecture can be found in Appendix C.

Thanks to the hierarchical architecture of UMAP, users can easily customize tasks through simple operations via top layers. Here we provide a detailed explanation of how each element of a task[8] is customized within UMAP.

**Agent Set.** Within the interface level of UMAP, the agent selection module enables users to specify the types, numbers, and associated teams of agents.

**State Space.** The global state is composed of the states of individual agents and the environmental state. Customization of the environmental state can be achieved by selecting different maps and modifying them along with related entities. The state of the agents can be customized through the *agent_init* function in the *advanced module layer* and the agent component module in the *interface layer*.

**Observation Space and Observation Function.** UMAP transmits global information from the UE side to the Python side, where the *make_obs* function in the *interface layer* is used to construct the agents' observations. Direct modification of this function allows for the customization of each agent's observation space and function. Moreover, modifying agents' properties, such as the observation range, can also change their observations. Additionally, UMAP supports more sophisticated agent observation simulation mechanisms, such as masking entities blocked by walls, which can be implemented through the agent perception module in the *advanced module layer*.

**Action Space.** UMAP supports continuous actions, discrete actions, and hybrid actions. Users can assign a built-in action set to each agent via the *agent_init* function in the *interface layer*. Furthermore, a deeper customization of agent actions can be achieved through the agent action-related modules in the *advanced module layer*.

**Transition Function.** Similar to the state space, the transition function in UMAP is comprised of local transitions of all agents and environmental transitions. The latter can be modified through map-related and entity-related modules. Local transitions of agents can be customized by modifying the *agent_init* function and the *step* function, or more deeply through the agent component modules and agent controller modules, such as agent kinematics.

**Reward Function.** UMAP constructs rewards using global information and an event system. Users can customize the agents' rewards by modifying the *make_reward* function, which supports team and individual rewards, as well as sparse and dense reward structures.

---

[7]Blueprint is a graphical programming language widely used in the UE editor.

[8]Excluding the discount factor, which can be easily specified on the algorithm side.

## 4 HMAP

To facilitate the deployment of algorithms for UMAP and fully utilize its capabilities, we also open-source our experimental framework HMAP. HMAP is a multi-agent experimental framework with decoupled Core-Task-Algorithm components. Currently, HMAP integrates environments such as UMAP, SMAC (Samvelyan et al., 2019), MPE (Mordatch & Abbeel, 2017), DCA (Fu et al., 2022), and OpenAI Gym (Brockman, 2016), and supports a wide range of algorithms. This includes rule-based algorithms (most of them are opponent policies for UMAP tasks), single-agent reinforcement learning algorithms like DQN (Mnih et al., 2015) and SAC (Haarnoja et al., 2018), as well as MARL algorithms such as MAPPO (Yu et al., 2022) and HAPPO (Zhong et al., 2024). Furthermore, HMAP is compatible with third-party frameworks, supporting all algorithms from PyMARL2 (Hu et al., 2021) and HARL (Zhong et al., 2024).

The unique feature of HMAP is its support for multi-team training. By thoroughly decoupling algorithms from tasks, HMAP employs its core as a "glue module", enabling any algorithm module to control teams within any task module. Moreover, the observations, actions, and reward data for each algorithm are processed separately and efficiently, ensuring that the policy executing and training for each team are independent. HMAP accommodates sequential and parallel updates of multiple team policy according to hardware performance variations, with the update sequence having no adverse impact on the effectiveness of algorithm training.

HMAP's highly modular design presents three key benefits. Firstly, it enables modification of script-based opponent policies, which are treated as algorithm modules, in contrast to SMAC and GRF where such policies are hardcoded and immutable. Secondly, it enables teams controlled by multiple algorithms to interact within the same scenario, facilitating the evolution and training of algorithms from different frameworks under the same task. Thirdly, it is user-friendly, as all experimental configurations based on HMAP can be implemented through a single JSON file. Upon completing the configuration, users can initiate the training task with just one line of code. More details of HMAP can be found in Appendix D.

## 5 SCENARIOS AND TASKS

UMAP includes a variety of basic scenarios for multi-agent systems, each of which is extensible and can be used to create numerous tasks. This section describes 4 primary scenarios, and 15 tasks applied in Section 6 generated from these scenarios. These primary scenarios incorporate both cooperative and competitive elements, including features as heterogeneous multi-agent, large-scale multi-agent, sparse team rewards, multi-team gaming, along with a sim-to-real demonstration.

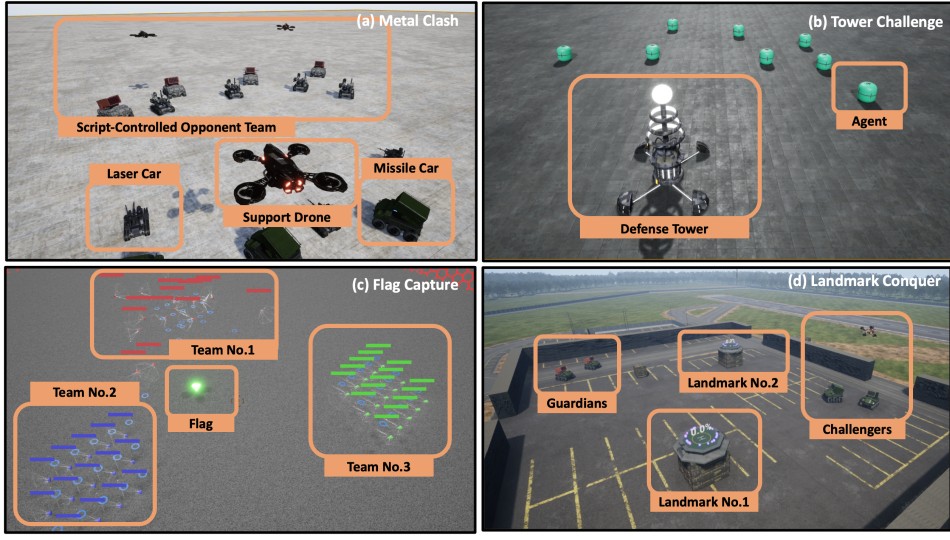

Figure 3: Four primary built-in scenarios of UMAP.

**Metal Clash.** This scenario is designed for heterogeneous and large-scale multi-agent tasks. As illustrated in Figure 3(a), the scenario involves a competition between two teams of agents. Each team can be controlled by either rule-based or learning-based algorithms. Metal Clash provides three types of basic agents: missile cars, laser cars, and support drones. The properties of each basic agent, such as maximum speed and health points (HP), are encapsulated as configurable parameters. Users can easily modify these parameters, creating a variety of heterogeneous agent types beyond the original three. Additionally, the number and types of agents in each team can be freely changed, altering the characteristics and difficulty of the tasks.

Based on this scenario, we develop a series of tasks with heterogeneous or large-scale features. In each task, the ally team, controlled by MARL algorithms, competes against an opponent team controlled by rule-based algorithms. Both teams have the same types and numbers of agents (in the following part, we only describe the composition of the ally team). We denote a task where the team consists of $x$ support drones, $y$ laser cars, and $z$ missile cars as *metal_clash_xsd_ylc_zmc*. Accordingly, we develop four heterogeneous tasks: *metal_clash_5sd_5lc*, *metal_clash_5sd_5mc*, *metal_clash_5lc_5mc*, and *metal_clash_2sd_4lc_4mc*.

For large-scale tasks, we develop two homogeneous tasks, *metal_clash_homo_50* and *metal_clash_homo_100*, which include 50 and 100 laser cars, respectively. In addition, there are two large-scale heterogeneous tasks, *metal_clash_hete_50* and *metal_clash_hete_100*. In *metal_clash_hete_50*, each team has 10 support drones, 20 laser cars, and 20 missile cars. In *metal_clash_hete_100*, the number of each type of agent is doubled compared to *metal_clash_hete_50*.

**Tower Challenge.** This scenario is designed for sparse team rewards in a multi-agent cooperative setting. As shown in Figure 3(b), it includes a defense tower and several agents. The goal of all the agents is to destroy the tower cooperatively. The tower's defenses cover a much larger area than any single agent can attack, making individual efforts ineffective. The entire team receives a positive reward only if the they destroy the tower, there are no rewards or penalties in other cases.

Users can adjust the difficulty by modifying the tower's HP and the number of agents. Based on this scenario, we design two tasks named *tower_challenge_easy* and *tower_challenge_hard*. Each task involves eight agents, with the harder task featuring a tower HP which is twice that of the easy one.

**Flag Capture.** This scenario is designed for multi-team gaming. As depicted in Figure 3(c), it involves several teams competing to capture a flag. The closest agent can pick up the flag, and their teammates must defend it from other teams. At the end of each episode, the team that held the flag the longest wins. Since all teams start with the same number of agents, capturing the flag first doesn't guarantee victory. Success requires balancing power and strategic cooperation among all teams. We develop 4 tasks based on this scenario. The first two tasks, *flag_capture_script* and *flag_capture_double_script*, correspond to two-team and three-team tasks, respectively. In each of these tasks, only one team is controlled by the tested MARL algorithm, while the remaining teams are controlled by scripts. Similarly, for the last two tasks, *flag_capture_mappo* and *flag_capture_double_mappo*, the script-based algorithms are replaced with MAPPO. Both the tested algorithm and the MAPPO algorithm(s) start training from scratch.

**Landmark Conquer.** This scenario is specifically developed to demonstrate the potential of sim-to-real transfer using UMAP. As illustrated in Figure 3(d), all agents and entities are derived from replicas of the physical environment described in Section 6.4. In this scenario, the challengers, consisting of two unmanned ground vehicles (UGVs) and one unmanned aerial vehicle (UAV), are tasked with capturing any landmark protected by guardians. Compared to the challengers, the guardians possess higher attack power and HP. The scenario includes several obstacles and walls, as well as two target locations. If the UAV remains above any landmark for a specified duration, the capture is considered successful, resulting in a victory for the challengers.

## 6 EXPERIMENTS

### 6.1 EXPERIMENTAL SETTING

Based on the 15 tasks developed in Section 5, we test 7 widely-used SOTA MARL algorithms. These include the actor-critic-based algorithms as MAPPO (Yu et al., 2022), HATRPO, and HAPPO (Zhong et al., 2024), as well as the value-based algorithms as QMIX (Rashid et al., 2020b),

QTRAN (Son et al., 2019), QPLEX (Wang et al., 2020), and WQMIX (Rashid et al., 2020a). To ensure a fair comparison, the main network structure of each algorithm is preserved uniform, and hyperparameters are standardized across all algorithms (refer to Appendix I for details).

The effectiveness of the training is tested after every 1280 episodes. The average win rates and rewards of the algorithms are calculated based on 512 episodes per test, across 5 or more random seeds. The results for the first 12 tasks are illustrated in Figure 4, where the lines represent the mean values and the shadowed areas indicate the 95% confidence interval. Table 2 details the performance of the 7 algorithms in *flag_capture_mappo* and *flag_capture_double_mappo*. Results for *landmark_conquer* are presented in Appendix G.

## 6.2 INTERPRETATION OF RESULT

**Heterogeneous Tasks**. The result plotted in Figure 4 reveals several trends. Apart from QPLEX, actor-critic-based algorithms generally outperform value-based algorithms. In actor-critic-based algorithms, MAPPO performs better, even being the best algorithm in the most difficult task, and HAPPO is weaker than MAPPO across all four tasks, which is different from previous research. In value-based algorithms, QPLEX is the best, which outperforms all actor-critic-based algorithms in *metal_clash_5mc_5lc*. However, it is discovered that the effectiveness of QPLEX significantly declines as the level of heterogeneity in the task increases. Furthermore, experiments without parameter sharing are conducted, and it has been found that actor-critic-based algorithms with parameter sharing outperform those without parameter sharing. Since the agent ID is already included in the observations, this enables differentiation among the trained policies.

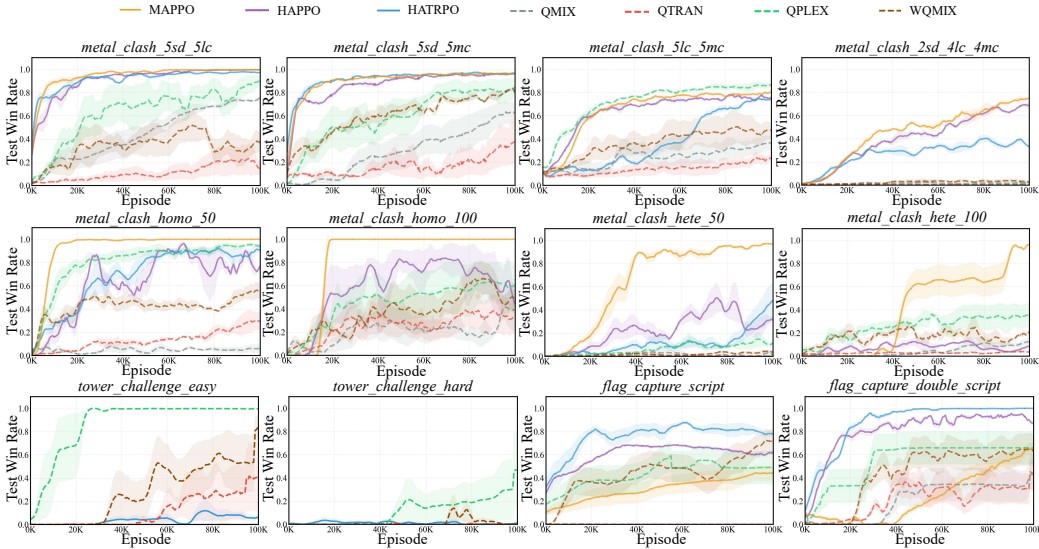

Figure 4: The comparison of test win rate for all evaluated algorithms across 12 tasks. The shadowed area depicts the 95% confidence interval.

**Large-Scale Tasks**. Similar to heterogeneous tasks, actor-critic-based algorithms still outperform value-based algorithms. MAPPO is the most outstanding algorithm due to its superior capability for parameter sharing, which is primarily reflected in its faster and more stable training performance. This advantage is particularly evident in *metal_clash_homo_100* and the highly heterogeneous *metal_clash_hete_100*, where MAPPO demonstrates a significant lead. For value-based algorithms, the performance of QPLEX is the best, but it also deteriorates rapidly with the increase in scale and heterogeneity. Furthermore, the training of HAPPO is very unstable, which may be related to its updating of policies in a random order. In tasks with 100 agents in the team, HATRPO freezes up and fails to produce results, because the computational burden of HATRPO is so large that it exceeds the computing capacity of the server. Apart from MAPPO and QPLEX in

*metal_clash_homo_50*, the performance of other algorithms is not satisfactory, urgently requiring more advanced algorithms.

**Sparse Team Reward Tasks**. In these tasks, value-based algorithms generally outperform actor-critic-based algorithms. In the *tower_challenge_easy* task, only QPLEX trains relatively quickly and stably. Other value-based algorithms require a larger number of episodes to exceed a win rate of 0.8 and do not perform well in the first 100,000 episodes. In *tower_challenge_hard*, some algorithms do not perform well, but as a actor-critic-based algorithm, HATRPO performs better than expected. The limitation of the performance of HATRPO in this task may lie in its inability to explore the entire space, thus failing to ensure monotonic improvement. Therefore, in *tower_challenge_hard*, there is an urgent need for more advanced algorithms.

**Multi-Team Gaming Tasks**. In the tasks where the engaging teams are driven by scripts, apart from the poor performance of MAPPO in all tasks, actor-critic-based algorithms are superior to value-based algorithms. Within actor-critic-based algorithms, HATRPO, as the algorithm with the most precise monotonic improvement, performs the best. It can stably learn the superior policies in both *flag_capture_script* and *flag_capture_double_script*. This indicates that in these tasks, computing only the first-order approximation or using clip clipping like HAPPO is not the optimal solution. Among value-based algorithms, QPLEX and WQMIX are the two best performing algorithms. Among them, QPLEX trains slightly faster, indicating that in simple tasks with fewer agents, QPLEX is the fastest learning algorithm among its value-based counterparts.

In *flag_capture_mappo*, actor-critic-based algorithms train relatively quickly and can achieve the high win rate within 50,000 episodes. On the contrary, value-based algorithms can achieve the high win rate only after 50,000 episodes. Except for QMIX, which performed poorly, the other algorithms performed well. In *flag_capture_double_mappo*, all actor-critic-based algorithms perform well. In value-based algorithms, only QPLEX can achieve the high win rate after a large number of episodes.

Table 2: The result of engaging with teams driven by MAPPO. The data represents the average win rate within the corresponding range of episodes.

| ALGORITHM | *flag_capture_mappo* | | | *flag_capture_double_mappo* | | |
|---|---|---|---|---|---|---|
| | $0k \sim 50k$ | $50k \sim 100k$ | $100k \sim 150k$ | $0k \sim 50k$ | $50k \sim 100k$ | $100k \sim 150k$ |
| MAPPO | $0.52 \pm 0.25$ | $0.56 \pm 0.09$ | $0.50 \pm 0.17$ | $0.71 \pm 0.12$ | $0.71 \pm 0.08$ | $0.78 \pm 0.17$ |
| HAPPO | $0.65 \pm 0.20$ | $0.67 \pm 0.17$ | $0.77 \pm 0.11$ | $0.67 \pm 0.08$ | $0.68 \pm 0.18$ | $0.71 \pm 0.17$ |
| HATRPO | $0.54 \pm 0.28$ | $0.67 \pm 0.16$ | $0.51 \pm 0.29$ | $0.61 \pm 0.26$ | $0.65 \pm 0.34$ | $0.77 \pm 0.12$ |
| QMIX | $0.11 \pm 0.07$ | $0.02 \pm 0.03$ | $0.01 \pm 0.02$ | $0.07 \pm 0.02$ | $0.65 \pm 0.34$ | $0.77 \pm 0.12$ |
| QTRAN | $0.71 \pm 0.06$ | $0.73 \pm 0.13$ | $0.78 \pm 0.06$ | $0.25 \pm 0.28$ | $0.17 \pm 0.24$ | $0.13 \pm 0.13$ |
| QPLEX | $0.66 \pm 0.21$ | $0.96 \pm 0.03$ | $0.97 \pm 0.02$ | $0.29 \pm 0.20$ | $0.73 \pm 0.09$ | $0.87 \pm 0.14$ |
| WQMIX | $0.39 \pm 0.18$ | $0.79 \pm 0.08$ | $0.76 \pm 0.19$ | $0.27 \pm 0.13$ | $0.07 \pm 0.05$ | $0.10 \pm 0.10$ |

### 6.3 EVERY TASK HAS ITS OWN SOTA ALGORITHM

Each task has its unique characteristics, which necessitate different suitable algorithms. In fact, no single algorithm currently dominates across all tasks, which implies the need for more advanced algorithms. Meanwhile, we summarize the above-mentioned algorithms as follows.

**MAPPO**. With strong parameter sharing capabilities, it is suitable for large-scale and simple tasks.

**HAPPO**. Compared to HATRPO, HAPPO has a lower computational burden but performs inconsistently in large-scale tasks. It performs better in simpler tasks such as multi-team tasks.

**HATRPO**. HATRPO has a significant computational burden, making it suitable for small-scale tasks, where it often performs better. Additionally, it tends to perform well in the early stages of multi-team tasks. However, it is difficult to run this algorithm for large-scale tasks.

**QMIX**. With the most classic mixing network, QMIX is suitable for the later stages of multi-team tasks, where it can stably suppress MAPPO.

**QTRAN**. The performance of QTRAN is relatively mediocre in the first 12 tasks. QTRAN exhibits better performance in *flag_capture_mappo*, with relatively stable training results. However, it still performs poorly in *flag_capture_double_mappo*.

**QPLEX**. QPLEX has strong performance capabilities, suitable for small-scale tasks with sparse team rewards.

**WQMIX**. As a relatively new algorithm, it outperforms QMIX in many tasks. It shows potential in tasks with sparse team rewards.

## 6.4 PHYSICAL EXPERIMENT

We conduct this experiment to demonstrate the potential of UMAP in bridging the sim-to-real gap. Firstly, we construct a real-world experimental setup, which consists of a motion capture system, a communication system, several autonomous UGVs and UAVs, and a number of physical entities. Subsequently, we develop the *landmark_conquer* scenario through UMAP, wherein the entities are proportionally replicated from the physical setup, and the kinematics of the unmanned units are also recreated. Ultimately, we develop an *algorithm-UMAP-hardware* framework, with details presented in Appendix G.

During the training phase, the algorithmic side, represented by HMAP, interact with UMAP to train policies within the simulated scenarios. In the execution phase, the physical system relay global information captured by the motion capture system and first-person view data from the vehicles' cameras to UMAP. UMAP then update its internal environment with this information and transmit the filtered observational data to HMAP. The algorithm within HMAP generate action commands based on these observations, which are conveyed to UMAP. UMAP execute virtual state transitions based on these commands, and concurrently transmit the decomposed action information to the real-world setup for execution by the autonomous vehicles/drones.

Figure 5 presents snapshots from both the virtual and the real-world scenarios. The experimental results indicate that the whole system can successfully replicate the policies of the multi-agent system from the virtual environment within the physical setup.

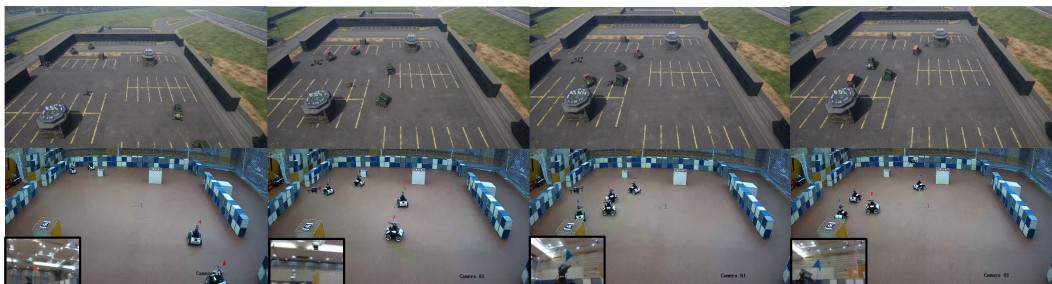

Figure 5: Snapshots from UMAP-simulated and real-world scenarios. The top four subfigures shows snapshots of multiple agents deploying well-trained policies only in virtual scenarios. The bottom four subfigures shows deployed policies in real-world scenarios at the same timesteps.

## 7 CONCLUSION AND FUTURE WORK

In this paper, we introduce UMAP, a powerful, highly extensible UE-based MARL simulation environment. Utilizing UMAP, we design a series of base tasks which include features such as heterogeneity, large scale, sparse team rewards, and multi-team. Additionally, we develop a multi-agent experimental framework compatible with UMAP, named HMAP. With the tasks developed on UMAP and the algorithm modules within HMAP, we provide a thorough report and discussion on several SOTA MARL algorithms, encompassing both value-based and actor-critic-based methods. Finally, we replicate a task in a real-world setting, demonstrating UMAP's potential to bridge virtual algorithms with real-world applications.

However, UMAP is not perfect. One limitation is that the sim-to-real demonstration so far is relatively simple and requires global real-world information to construct pretended local information. In the future, we plan to develop a comprehensive, plug-and-play sim-to-real toolkit based on UMAP. This toolkit will help map real-world requirements into UMAP's virtual environment, thereby advancing the practical application of MARL to the next level.

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

## A    OPEN SOURCE STATEMENT

We are willing to open source all the components of UMAP and HMAP to benefit the MARL community. Due to the constraints of the paper review rules and the file size limitations for supplementary materials, we temporarily include only the most crucial components in the supplementary material. These components comprise a lightweight version of the HMAP framework and the Python interface for UMAP.

In fact, we have a detailed plan for open-sourcing, which will be executed after the review process. The open-source plan is as follows:

1. **Regarding code environment configuration:** We will release a Docker image supporting UMAP and HMAP services on Docker Hub. This image will include the HMAP framework, a default version of UMAP's compiled binary files, and a series of environment configurations.

2. **Regarding UMAP:** We will publish UMAP's usage tutorials and one-click deployment scripts on GitHub. These scripts facilitate the compilation of rendering/training-only binary files for various platforms and automate the downloading of large files. The Unreal project and the modified Unreal Engine of UMAP will be available on a cloud drive, accessible for automatic download via Python scripts.

3. **Regarding HMAP:** We will publish HMAP's usage tutorials and its entire content to GitHub. This content includes the core of HMAP, wrappers for all supported environments, built-in algorithms, and algorithms from third-party frameworks.

4. **Future Plans for Open Source Work:** We will continue to maintain all GitHub repositories, develop new scenarios, incorporate more algorithms from third-party frameworks, and develop sim-to-real related toolkits.

## B    RELATED WORK

The simulation environments for MARL can be broadly categorized into two types: those with physics engines and those without. Here, physics engines refer to a suite of tools capable of simulating the physical laws inherent in real-world tasks (Templet, 2021). Given that game engines also aim at reincarnating the real-world elements into the digital world (Vohera et al., 2021), environments leveraging game engines are classified under the physics engine category.

Among the environments without physics engines, MPE (Mordatch & Abbeel, 2017) utilizes a simple rule-based particle world to simulate multi-agent tasks such as predator-prey and cooperative navigation. MAgent (Zheng et al., 2018), grounded in a grid world, facilitates simulations involving the aggregation and combat of pixel-block agents, notable for its ability to support large-scale multi-agent settings. The two environments mentioned above are based on the state transition laws of particle worlds and particle interactions. Although they are completely open-source and their task elements are relatively easy to modify, their scenarios are overly simplistic and lack realism.

Hanabi (Bard et al., 2020) provides a multiplayer card game scenario, which is commonly used in MARL research based on opponent modeling. However, the overly narrow theme prevents it from further simulating tasks involving heterogeneity, large scale, and mixed strategies. Neural MMO (Suarez et al., 2021) is developed in a 3D grid world derived from massively multiplayer online games, supporting large-scale multi-agent simulations over long time horizons.

Gobigger (Zhang et al., 2022), based on a ball world concept, stands out for enabling simulations involving collaboration and competition among multiple teams. However, similar to MPE and MAgent, their particle-based 2D environments fall significantly short of simulating the real-world complexities of 3D scens.

JaxMARL (Rutherford et al., 2024) integrates numerous MARL environments together and has re-implemented these environments using JAX technology, enabling them to support efficient, GPU-based parallel computing. However, to support pure GPU parallelism, some environments in JAX-MARL have lost their original CPU-based underlying physical engines. Moreover, as a collection of environments that integrates multiple basic environments, it does not support multi-team multi-

algorithm training, nor does it support controllable time-flow simulation and cross-platform real-time rendering.

As for the environments with physics engines, GRF (Kurach et al., 2020) is built upon the Gameplay-Football simulator (Schuiling, 2017), creating a highly realistic football match setting that allows agents to simulate the behaviors of human players. However, it does not support large-scale scenarios, multi-team training, and mixed multi-agent gameplay. Moreover, although its environment interface and underlying engine are open-source, the underlying engine is not suitable.

SMAC (Samvelyan et al., 2019) and SMACv2 (Ellis et al., 2022) are developed based on the popular video game StarCraft II, constructing a multi-agent micromanagement environment where each agent controls individual units to complete adversarial tasks. Despite their widespread use, the fact that their underlying games and engines are not fully open-source limits further expansion, confining their built-in tasks to battle-type game scenarios only.

Hide-and-Seek (Baker et al., 2019) has set up a series of multi-agent curriculum learning scenarios, such as hide and seek, based on a 3D engine. However, its theme is too singular, making it impossible to simulate tasks involving heterogeneity, large scale, multiple teams, etc., and it does not allow for customization of all task elements. Hok3v3 (Liu et al., 2023), specifically designed for heterogeneous multi-agent tasks, is based on the Honor of Kings engine, with agent action spaces consistent with those of human players engaging in the real game. However, it only supports heterogeneous multi-agent scenarios (3VS3) and does not have an open-source underlying game and related engine.

MAMuJoCo (Peng et al., 2021a) is developed using the Mujoco physics engine (Todorov et al., 2012), where multiple agents each control different joints to collaboratively manage the movements of a single robot. However, all of the multi-agent scenarios are fully cooperative and do not support large-scale multi-agent tasks.

Marathon Environment (Booth & Booth, 2019) is developed using the Unity3D engine, supporting multiple agents learning complex movements such as running and backflipping. The built-in tasks are relatively simple and are unable to simulate large-scale, multi-team, and mixed multi-agent gameplay tasks. Moreover, its underlying engine, Unity3D, is not fully open-source, thus preventing comprehensive modifications from the bottom to the top layer.

It is evident that environments without physics engines are adept at simulating challenging tasks designed to push the limits of existing algorithms. In contrast, environments equipped with physics engines offer greater potential for real-world applications but are constrained in terms of academic flexibility. Our goal is to develop an environment that not only has practical application potential but also fully leverages scalability, ultimately leading to the creation of UMAP.

## C  UMAP Details

### C.1  Architecture of UMAP

UMAP utilizes a hierarchical design that consists of five layers, all of which are open source. As shown in Figure 2, the first layer of UMAP is the native part of the Unreal Engine, including the physics engine, rendering engine, AI engine, and a range of 3D assets. We build the entire UMAP based on the open-source version of UE, making modifications to some of the native modules. For instance, the original AI detection system for agents in UE was very inefficient in large-scale scenes. UMAP optimizes the detection of multiple entities by incorporating tensor operations and eliminating redundant checks.

The second layer of UMAP comprises the underlying systems and programming specification, all implemented in C++. The time control system and task system in this layer ensure the correct initiation and termination of simulation episodes, guaranteeing the precision of simulation time steps and the reproducibility of experimental results. Other components of this layer define the specification for all base classes, communication, and debugging within UMAP.

The third layer consists of three fundamental classes implemented using Blueprints. The agent class defines all entities that can be controlled by algorithms, while the entity actor corresponds to all environmental entities that do not make decisions. Classes derived from these two form all the

entity elements within a task scenario. The abstract class acts as a bridge, connecting the underlying systems to the highest Python-based layer, facilitating communication, debugging, action updates, and observation feedback.

The fourth layer of UMAP consists of advanced functional modules, implemented using Blueprints. These modules allow for the modification of various attributes of agents, including appearance, perception, action sets, movement, and navigation, enabling the development of diverse types of agents. Moreover, leveraging the abundant resources in the Unreal community, the map construction module facilitates the creation of new maps and even the importation of real-world maps. The entity construction module aids in developing complex environmental entities, such as altering the kinematic model of missiles launched by drones.

The fifth layer serves as the interface for interaction between UMAP and algorithms, all implemented in Python. This interface adheres to the gym (Brockman, 2016) specification, encompassing basic functions like reset, step, and done, and supports the customization of agent-level observation and reward functions. Attributes such as agent size, initial position, detection range, and health are directly encapsulated within the agent initialization function, allowing for easy modification. As shown in Figure 6, the selection of agents, tasks, and maps in UMAP are independent. Users can customize the types, numbers, and teams of agents in a task and switch maps flexibly.

From the perspective of designing and utilizing a MARL simulation environment, users need only focus on the fourth and fifth layers of UMAP. In most cases, users can directly customize MARL tasks by modifying the built-in scenarios and agent parameters through the fifth layer. If there is a need to develop new scenarios or further develop existing ones, users can also easily develop through the graphical programming approach provided in the fourth layer. UMAP's hierarchical design significantly reduces the burden of customizing tasks.

## C.2    TIME IN UMAP

Time is the most important factor in simulations. There are two different type of time in UMAP:

1. Real World Time $t_{real}$. The actual time of our world.

2. Simulation Time $t_{sim}$. The time in the simulated virtual world.

It is inevitable that simulation speed (from the perspective of $t_{real}$) will be influenced by factors such as CPU frequency, GPU performance, policy neural network size, machine workload, etc. As a result, UMAP decouples simulation time flow therefore has achieved flexible control of simulation time

1. UMAP allows researchers to slow down simulation time by setting a time dilation factor, extending a second in the simulation multiple times to render details of agents in slow motion.

2. UMAP allows researchers to accelerate simulation time by setting the same time dilation factor (before reaching the hardware limitation). Gathering large amount of samples is necessary in most RL tasks. Accelerating computation is the primary ways to achieve this goal.

UMAP guarantees that the simulation results will not be influenced by time dilation factor, hardware or workload. For instance, as long as the random seed remains identical, same agent trajectories are expected: 1) regardless of whether we choose to enable GPU to accelerate neural network computation. 2) regardless of whether we choose to simulate agents slowly or rapidly by setting different time dilation factors.

There are three global time-related settings to adjust in UMAP.

**Decision time interval.** From the perspective of agents in the simulated environment, agents will have a chance to act once every Alternatively, $t_{sim}^{step}$ is also the time interval between each RL step. $t_{sim}^{step}$ is usually a short period with a default value 0.5s. Nevertheless, for tasks such as flights that last hours in a episode, $t_{sim}^{step}$ should be increased accordingly.

$t_{\text{sim}}^{\text{step}}$ does NOT has directly relationship with how long a RL step will actually take in the real world. More specifically, a team can take as long as necessary to compute the next-step action after receiving observation, meanwhile the simulation time flow freezes until all teams have committed agent actions. In extreme situations, algorithms can spend hours to update large policy networks and the simulated agents will not be influenced by this delay.

**Baseline Frame Rate.** Baseline Frame Rate $t_{\text{sim}}^{\text{fr}}$ determines how many frames to compute for each simulation second in UMAP. As an example, when $t_{\text{sim}}^{\text{fr}} = 30$, the simulation will proceed (tick) $\frac{1}{30}$s after each frame. Important computation such as collision detection and agent dynamic update are performed in each of these frames. As an example, let $t_{\text{sim}}^{\text{step}} = 0.5$ and $t_{\text{sim}}^{\text{fr}} = 30$, under this circumstance 15 ticks will be performed between each RL step. Similarly, $t_{\text{sim}}^{\text{fr}}$ does NOT have direct relationship with the real world time flow.

**Time Dilation Factor.** In UMAP, Time Dilation Factor $t_{\text{real}}^{\text{df}}$ is the sole bridge between simulation time flow and real world time flow. In reinforcement learning, there are three typical cases that involves the control of time in simulation:

1. Task Development and Evaluation. In this case, it is demanded that simulation time flows at a normal speed to observe the interaction of agents. A dilation factor $t_{\text{real}}^{\text{df}} \approx 1$ will synchronize simulation time flow with the real world time flow.

2. Slow Motion. In this case, it is required that the simulation runs slowly to allow human observers to diagnose issues in multi-agent cooperation. Changing the dilation factor $t_{\text{real}}^{\text{df}} < 1$ will slow down the simulated world accordingly.

3. Training. In this case, it is demanded that simulation runs as fast as possible to collect training data. UMAP will attempt to accelerate the simulation until reaching the $t_{\text{real}}^{\text{df}}$ threshold. If not possible due to hardware, the simulation will still proceed at the fastest possible simulation speed.

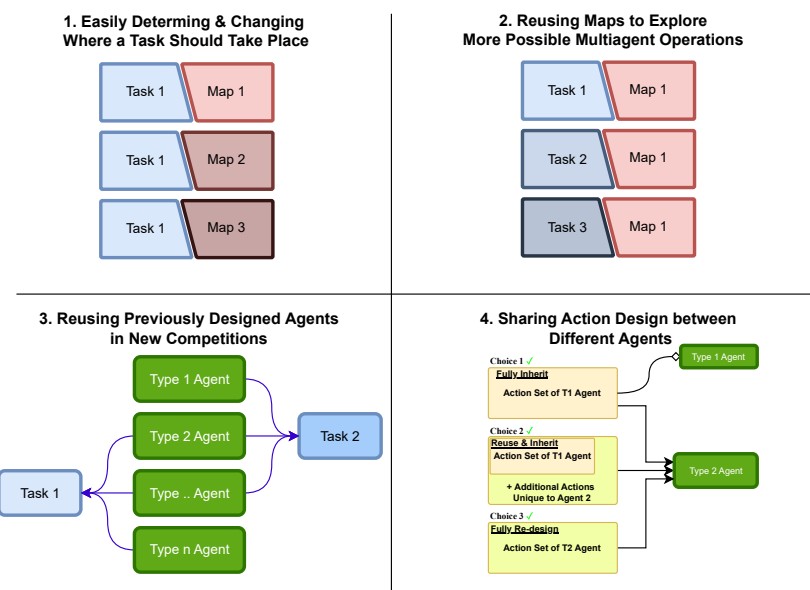

Figure 6: One of the advantages of UMAP framework is the isolation of maps, task and agents, making it possible to reusing existing modules to develop new environment for Reinforcement Learning studies.

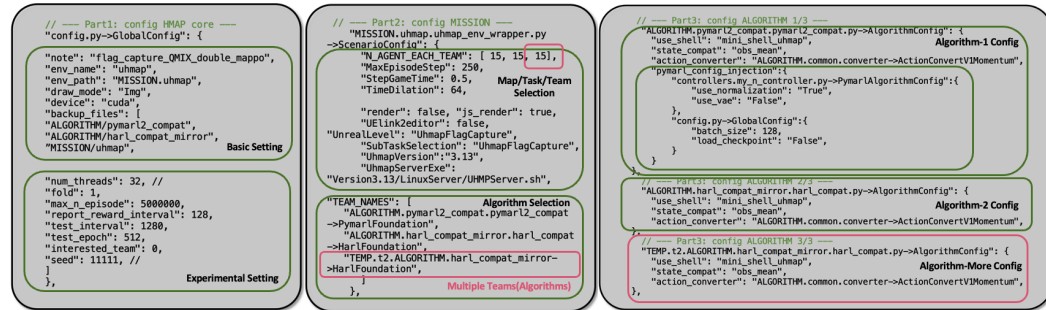

Figure 7: Parameter configuration of HMAP. This is an example under the *flag_capture_double_mappo* task using the QMIX algorithm. To facilitate multi-team training, users only need to add teams and designate their respective algorithms in the mission config, and append the corresponding algorithmic parameters in the Algorithm config.

# D HMAP DETAILS

The utilization of HMAP is straightforward, necessitating only a Docker container, a configuration of parameters, and execution of a single command to deploy a specified algorithm into a designated scenario. The parameter configuration files of HMAP exemplify its modular design, including three categories: core config, mission config, and algorithm config. Figure 7 illustrates a configuration file for the QMIX algorithm under the *flag_capture_double_mappo* task. **Core config** comprises basic settings and experimental settings, where the former allows for the specification of file and path for experiment storage, and the latter includes parameters relevant to the experiment such as the number of parallel task environments, testing intervals, and random seeds. **Mission config** includes selections for the simulation environment and deployed algorithm. Upon selecting UMAP as the simulation environment, users can make further selections regarding maps, tasks, and teams, as well as choose between training, rendering, or a mode that combines both training and real-time rendering. **Algorithm config** is composed of the algorithmic parameters set for each team.

With HMAP, users can conveniently specify the number of teams and freely assign algorithms to each team. For instance, Figure 7 demonstrates the setting of three teams, where Team-1 is assigned QMIX from the PyMARL2 framework, and Team-2 and Team-3 are designated MAPPO from the HARL framework. HMAP allows multiple teams to utilize the same algorithm module without affecting the normal construction of buffers and network updates. It is achieved by adding a prefix keyword like "TEMP.t2" to the additional configurations of the same algorithm. Theoretically, as long as computational resources are sufficient, UMAP and HMAP can support an arbitrary number of teams, each allocated with different algorithms in a same scenario, with the updates of different algorithms not interfering with each other.

# E SCENARIOS DETAILS

**Metal Clash.** This scenario is designed for heterogeneous multi-agent tasks and large-scale multi-agent tasks. Within this scenario, an ally team need to confront an enemy team controlled by built-in scripts or MARL policies. The objective of the ally team is to eliminate as many enemy agents as possible while preserving more ally agents.

Metal Clash offers three types of basic agents: missile cars (for ground and air attacks), laser cars (for ground attacks), and support drones (for attacks and supports). Missile cars can attack ground or aerial units with missiles and have a long range, but they move slowly. Laser cars excel at close-range combat, using lasers to damage ground units. Support drones, as aerial units, have a faster movement speed and can restore the health points of allied missile cars and laser cars. They can also attack opponents with smaller firepower but have lower HP.

The physical attributes of each basic agent, such as movement speed, size, HP, observation radius, etc., are exposed as configurable parameters at the task construction interface. Users can conveniently modify all the parameters, thereby creating a variety of heterogeneous agent types that far exceed the original three. Agents can sense their neighborhood allies and enemies within the perception range, and the information of perceived agents is concatenated in the observation space. Due to the height advantage, the flying agents have a much larger perception range than ground agents. Agents have rich options of actions. Besides the idle and moving actions, agents can choose a patrol-moving action to search enemies, select visible opponents to attack, or toggle their micromanagement strategy (such as whether agents are allowed to peruse opponents after receiving future attack action). Furthermore, users can freely designate the types and corresponding numbers of agents in both allied and enemy teams, thus controlling the nature and difficulty of the task.

- *Observation*

Three types of agents are intentionally designed differently to reflect the heterogeneity of this scenario. We define the distance unit of the unreal engine as $u$. Missile car have a maximum movement speed of 500u per second, an attack power of 1, an attack range of 1000u, and 150 HP. Laser cars have a maximum movement speed of 800u per second, an attack power of 1, an attack range of 500u, and 100 HP. Support drones have a maximum movement speed of 1000u per second, an attack power of 1/6, an attack range of 1700u, and 50 HP. The observation capabilities of three agents are shown in Table 3. The observation structure refers to the composition of what an agent observes, where the number 1 represents its own observation, and the subsequent two numbers indicate the maximum number of allied agents and enemy agents that can be observed. For example, missile cars have an observation range of 2500u, and their observation structure is [1,8,8]. This indicates that missile cars can observe information about up to 10 ally agents and 10 foe agents within a range of 2500u. Additionally, the information for each observed agent is a 20-dimensional vector, with vectors for invalid entities filled with zeros. Therefore, the observation dimension for the missile vehicle is (17)*20.

Table 3: Observation Capabilities of three base agents in Metal Clash.

| Agent | Observation Range | Observation Structure | Observation Dimension |
|---|---|---|---|
| Missile Car | 2500u | [1,8,8] | (17)*20 |
| Laser Car | 2000u | [1,5,5] | (11)*20 |
| Support Drone | 2500u | [1,10,10] | (21)*23 |

- *Action*

All three types of agents have nine common actions: moving in four directions, staying still, targeting foe agents within the defense circle, fleeing, etc. On this basis, each type of agent can perform special actions. For example, support drones can choose to restore the health of ally agents within their support range or choose to attack foe agents. Missile cars can choose to attack all units, while laser cars can only choose to attack ground units.

- *Reward*

Regarding reward settings, when an agent from our team or the enemy team is destroyed, the entire team receives a penalty of 0.05 and a reward of 0.1. At the end of an episode, the team with the higher total remaining HP wins, receiving a reward of 1.0, while the losing team receives a penalty of -1.0. In the event of a tie, both teams receive a penalty of -1.0.

**Tower Challenge.** This scenario consists of a defense tower and several agents. Users can adjust the task's difficulty by altering the tower's defensive capabilities and the number of agents involved. In the most challenging cases, agents must form a precise formation beyond the tower's defense range and launch a swift, simultaneous attack to just manage to destroy the tower.

Regarding reward settings, the entire team receives a positive reward only if the agents successfully destroy the tower; there are no rewards or penalties in other cases. In this experiment, We develop a simple cooperative task with the defense tower's HP set to 400 and a difficult cooperative task with the defense tower's HP set to 800. Similar to the Metal Clash scenario, both the tower and the agents have spherical perceptual space centered around themselves. Agents can choose from idle, move to a certain direction, or attack in their action space.

- *Observation*

In Tower Challenge, each agent has the same observation space. They can observe information such as the ID, HP, position, and maximum speed of themselves and the position of the tower.

- *Action*

In Tower Challenge, agents have six possible actions to choose: moving in four directions, maintaining the current action, and staying still. Additionally, each agent has an action to collide with the tower.

- *Reward*

This scenario is set up for sparse team rewards. Rewards are given only when all agents cooperate to destroy the defense tower. Specifically, agents can receive a reward of value 1 only when they defeat the defense tower.

**Flag Capture.** This scenario allows for competition among more than two teams. At the end of an episode, only the team that holds the flag for the longest duration wins. Since each team begins with an equal number of agents, the first team to capture the flag does not guarantee victory, as teams must carefully consider the balance of power and strategic play among multiple teams to receive the most rewards. In this scenario, agents are not equipped with weapons and cannot eliminate other agents. Consequently, agents do not have attack actions. Moreover, the agent's perceptual space is conical rather than spherical.

- *Observation*

In Tower Challenge, each agent has the same observation space. They can observe information such as the ID, HP, position, and maximum speed of ally or foe agents within the observation range.

- *Action*

Each agent has a constant speed. In the two-dimensional plane, there are eight discrete actions to choose from, each representing a direction spaced 45 degrees apart. When the team is close enough to the flag, the agent nearest to the flag will pick it up. To prevent other teams from approaching and capturing the flag, it is necessary to target the agent that are near the flag.

- *Reward*

When a flag is picked up by an agent, the team to which the agent belongs receives a reward of 0.005. At the end of the episode, the team that has held the flag for the longest time will receive a reward of 1.0.

**Landmark Conquer.** This scenario is specifically developed to demonstrate the potential of sim2real transfer using UMAP. It features a straightforward structure and components for easy replication and setup. Within this environment, a MARL algorithm must control an offensive unit consisting of two unmanned ground vehicles (UGVs) and one unmanned aerial vehicle (UAV) to attempt to capture a strategic area defended by two UGVs with double the attack power and HP of the offensive UGVs. The offensive UAV must seize control of the target area under the cover of the UGVs. The scene includes various obstacles and walls, along with two strategic points. The offensive team is deemed victorious if the UAV hovers above any strategic point undisturbed for 10 seconds. Failure occurs if the offensive team is eliminated or fails to capture any strategic point by the end of an episode.

- *Observation*

In this scenario, agents can perceive the location and status of the target area regardless of the distance, yet can only sense and attack opponents within agents' perception range.

- *Action*

All three types of agents have nine common actions: moving in four directions, staying still, targeting foe agents within the defense circle, fleeing, etc. On this basis, each type of agent can perform special actions. For example, support drones can choose to restore the health of ally agents within their support range or choose to target foe agents.

- *Reward*

Regarding reward settings, when an agent from our team or the enemy team is destroyed, the entire team receives a penalty of 0.05 and a reward of 0.1. At the end of an episode, the team with the higher total remaining HP wins, receiving a reward of 1.0, while the losing team receives a penalty of -1.0. In the event of a tie, both teams receive a penalty of -1.0.

## F  UMAP'S EFFICIENCY AND COMPUTATIONAL RESOURCE CONSUMPTION

As a simulation environment based on a 3D physical engine, UMAP boasts high simulation efficiency. It is well-designed to adapt to and fully utilize various types of computing resources. UMAP can be deployed on computing systems entirely devoid of GPUs for algorithm training and supports the full utilization of uneven computing resources. It can operate in a single-threaded manner as well as support multiple parallel environments. Additionally, with the feature of time dilation factors, UMAP can not only improve simulation efficiency by increasing the number of parallel processes but also control the simulation speed of each process to make full use of computing resources (using more CPU utilization under the same memory and GPU memory), a functionality not available in other simulation platforms.

In this section, to verify UMAP's efficiency and adaptability to various computing resources, we conducted a series of experiments on UMAP's efficiency index and resource consumption indices. The efficiency index adopted was FPS, i.e., the number of virtual timesteps run in a real second; the resource consumption indices included CPU utilization, memory occupancy, and GPU memory occupancy. All experiments were conducted on a Linux server equipped with an AMD7742 CPU (maximum frequency 2.25GHz) and NVIDIA RTX3090 GPUs. To ensure fairness, all experiments tested the QMIX algorithm on the *metal_clash_5sd_5mc* task. The data points for all indices were obtained by averaging the results of five experiments. At the beginning of each experiment, the server was maintained in an idle state, executing only the essential system processes.

Figure X shows the indices under a fixed number of parallel environments at 8, with varying time dilation factors. Figure Y shows the indices with a time dilation factor of 32, under varying numbers of parallel environments. From these two figures, the following conclusions can be drawn:

1. With a constant number of parallel environments, FPS and CPU utilization are roughly proportional to the time dilation factor, but this proportional relationship degrades into a positive correlation when the time dilation factor reaches a certain threshold (limited by the CPU's clock speed).

2. With a constant number of parallel environments, changing the time dilation factor almost does not affect memory occupancy and GPU memory occupancy.

3. With a constant time dilation factor, CPU utilization is roughly linearly related to the number of parallel environments, while FPS is roughly logarithmically related; memory and GPU memory occupancy are positively correlated with the number of parallel environments.

The above conclusions mean that under limited memory resources, training efficiency can be improved by increasing the time dilation factor to fully utilize CPU resources; similarly, under limited CPU computing resources, reducing the time dilation factor and increasing the number of processes can avoid the waste of computing resources.

In fact, when the number of processes is 8 and the time dilation factor is 32, training 1024 episodes on the *metal_clash_5sd_5mc* task takes less than 2 minutes. This means that under such parameter settings, this server can simultaneously support 50 such tasks (each with 20 agents) and complete all training tasks (100k episodes) within 3 hours. In special cases, the number of parallel processes can be further increased to improve training efficiency. When the number of processes reaches 128 and the time dilation factor is set to 32, the FPS can reach 1000+, and the training task can be completed in about an hour.

It is important to emphasize that FPS here counts the number of virtual UMAP timesteps per real second. Considering this is a simulation of 20 agents, and each timestep in UMAP undergoes 1280 frames of calculations for environmental dynamics and kinematics to maintain fine state transitions (details in Appendix C.2), this is already highly efficient computation.

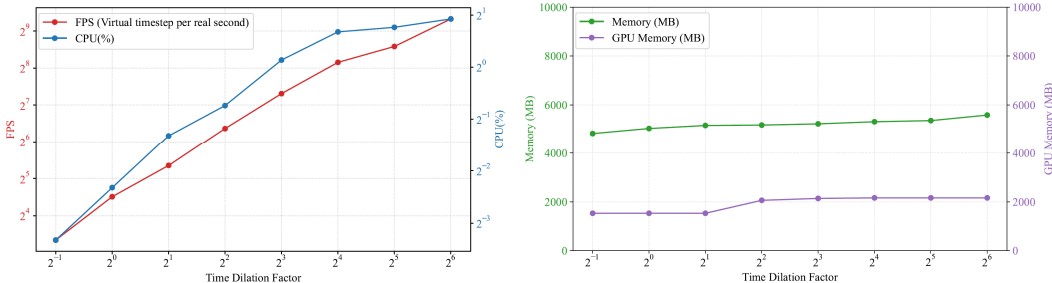

Figure 8: Efficiency index and resource consumption indices of running the QMIX algorithm with different time dilation factors under 8 parallel environments in the *metal_clash_5sd_5mc* task.

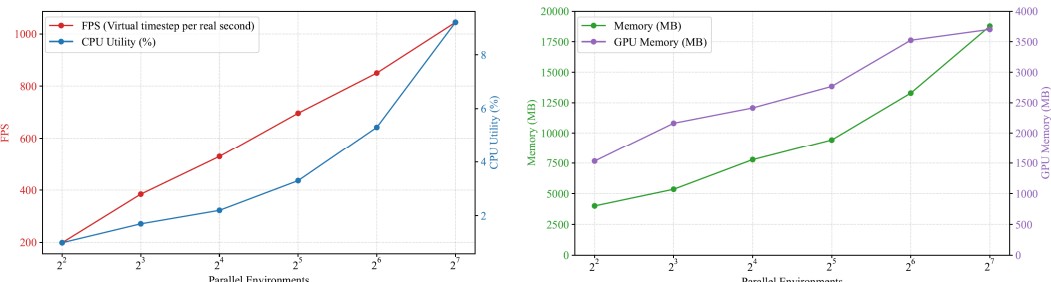

Figure 9: Efficiency index and resource consumption indices of running the QMIX algorithm with different numbers of parallel environments at a time dilation factor of 32 in the *metal_clash_5sd_5mc* task.

## G PHYSICAL EXPERIMENT DETAILS

As shown in Figure 11, the overall framework of the physical experiment includes three components: the algorithm side represented by HMAP, the virtual environment side represented by UMAP, and the hardware-based real environment side. During the training phase, HMAP and UMAP communicate through the TCP protocol, exchanging observations and action information of the environment, completing the training tasks on the same host server/computer. During the executing phase, UMAP needs to maintain communication with not only HMAP but also with the communication system in the real environment side through the TCP protocol, transmitting global observation information and decoded action information. In addition to the communication system, the real environment side also includes an action capture system, several UAVs and UGVs, landmarks and obstacles, and a host computer. The motion capture system transmits global information (the position, speed of all entities) to the host computer through a wired network, which receives local observation information (such as the first-person view from cameras) from UGVs and UAVs through a wireless communication module and sends commands to them.

The UGVs and UAVs in the real environment have autonomous planning and control capabilities. They can receive the information of target position or target speed from the communication module and complete commands through two-dimensional and three-dimensional PID control. UMAP also replicates their PID kinematics. UGVs are also equipped with cameras, which send the first-person view information to UMAP. UMAP simulates their viewpoints, combined with the global information from the motion capture system, to create simulated observation information under partially observable conditions for HMAP. In the simulated environment training, the simulated UAVs also have limited viewpoints, being able to observe entity information only within a specified range and angle.

It is worth mentioning that *landmark_conquer* itself is also a MARL task, on which we test the performance of 7 algorithms, as shown in Figure 10. In the physical experiment, we transfer the policy of the MAPPO algorithm to the real environment after 100k training episodes.

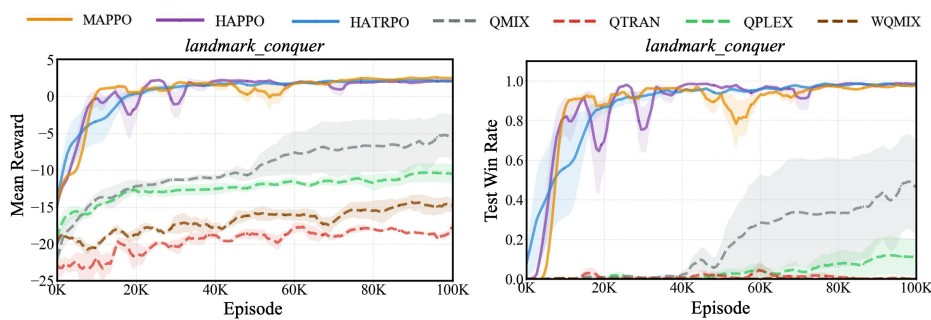

Figure 10: Results of the *landmark_conquer* task.

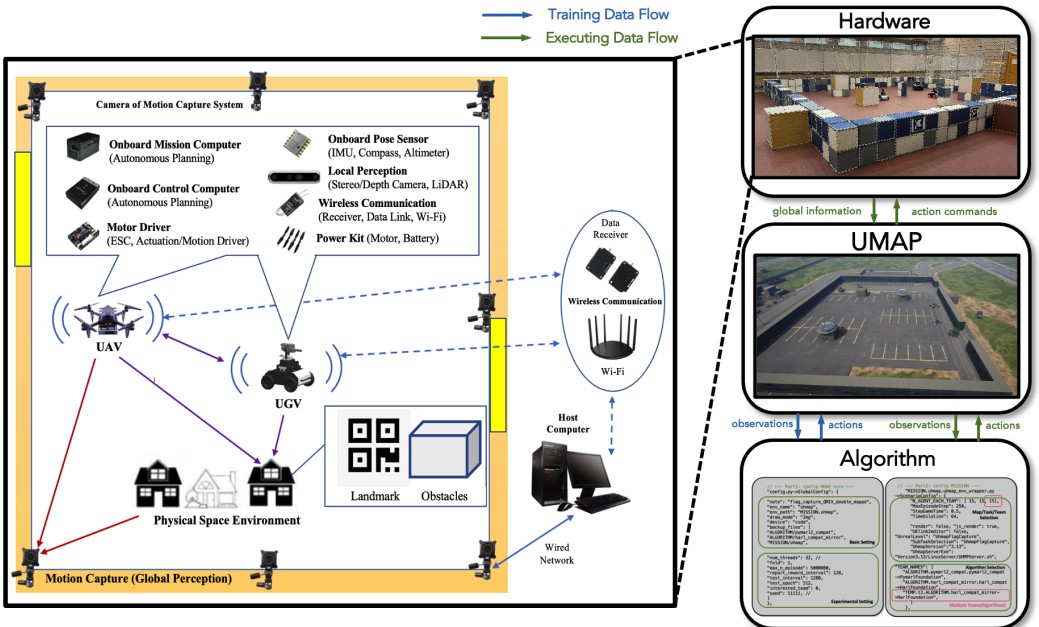

Figure 11: The algorithm-UMAP-hardware framework.

## H CHECKLIST OF ETHICS

We have finished a checklist to facilitate discussions on the ethical considerations of artificial intelligence involved in our work. This checklist (Kaffee et al., 2023) addresses the potential impacts of various artefacts in the field of artificial intelligence.

**C1 Did you explicitly outline the intended use of scientific artefacts you create?**

Yes. The scientific artefacts we have created are UMAP and HMAP. The former is a extensible simulation environment developed based on the Unreal Engine, designed with a layered architecture to enable users to conveniently develop various realistic 3D multi-agent simulation environments. The latter is an experimental framework that is highly compatible with UMAP, characterized by its support for multi-team multi-algorithm training, and compatibility with existing classic simulation environments and algorithms from third-party frameworks. The purpose of developing UMAP and HMAP is to enable users to develop simulation environments that meet their needs (including sim-to-real transfer and new research ideas) in the field of MARL, and to rapidly deploy algorithms to validate ideas, thereby promoting the development of the MARL field.

**C2 Can any scientific artefacts you create be used for surveillance by companies or governmental institutions?**

No. As a simulation environment and experimental framework in the MARL field, UMAP and HMAP are unrelated to surveillance by companies or governmental institutions.

**C3 Can any scientific artefacts you create be used for military application?**

The motivation and details of creating UMAP and HMAP are unrelated to any military applications. However, it must be emphasized that although the design inspirations, virtual materials, and physical materials used in this work are unrelated to military applications, there is a risk if our work is applied to MARL policy training for military purposes. Therefore, on one hand, we call on the open-source community to strengthen the regulation of military application materials and urge users to refrain from using UMAP for military purposes. On the other hand, we also plan to set up keyword detection within the UE side, so that users with impure motives designing military application-related scenarios will not be able to use the functions of UMAP.

**C4 Can any scientific artefacts you create be used to harm or oppress any and particularly marginalised groups of society?**

The motivation and details of creating UMAP and HMAP are unrelated to harming or oppressing any particularly marginalized groups of society. In fact, our environment and experimental framework are suitable for users under various computing resources, and are compatible with various system platforms.

**C5 Can any scientific artefacts you create be used to intentionally manipulate, such as spread disinformation or polarise people?**

The motivation and details of creating UMAP and HMAP are unrelated to intentional manipulation, such as spreading disinformation or polarising people. However, it must be emphasized that although the experiments and demonstrations based on UMAP are unrelated to this. If the simulation environments developed using UMAP can be used to generate realistic false scenarios, there is a risk of being maliciously used to create and spread false information. Therefore, we call on the open-source community to participate in regulation, establish a reporting mechanism, and we will add educational materials for users in the usage tutorials, emphasizing the ethical responsibility of using simulation environments and raising users' ethical awareness.

**C6 Did you access your institution's or other available resources to ensure limiting the misuse of your research?**

Yes, we have accessed our institution to ensure limiting the misuse of our research, including but not limited to the promotion, use, and modification of this work.

**C7 have you been provided by your institution with ethics training that covered potential mis-use of your research?**

Yes, we are confident that our institution has provided sufficient ethics training.

**C8 Were the scientific artefacts you created reviewed for dual use and approved by your institution's ethics board?**

Yes, the scientific artefacts we created have been reviewed for dual use and approved by our institution's ethics board.

## I HYPERPARAMETER DETAILS

In this part, the common hyperparameters used for algorithms and tasks are described. We present the hyperparameters used for actor-critic-based algorithms in Table 4 and for value-based algorithms in Table 5 across all tasks. Other unspecified hyperparameters of algorithms remain at their default settings. The hyperparameters used for tasks are shown in Table 6.

Table 4: Common hyperparameters used for MAPPO, HAPPO, and HATRPO in UMAP.

|  | MAPPO | HAPPO | HATRPO |
|---|---|---|---|
| share parameter | True | True | True |
| hidden sizes | 128 | 128 | 128 |
| use feature normalization | True | True | True |
| use naive recurrent policy | False | False | False |
| actor learning rate | 0.001 | 0.001 | 0.001 |
| critic learning rate | 0.0005 | 0.0005 | 0.0005 |
| eps of optimizer | 0.00001 | 0.00001 | 0.00001 |
| weight decay | 0 | 0 | 0 |
| clip parameter | 0.2 | 0.2 | 0.2 |
| entropy coefficient | 0.01 | 0.01 | 0.01 |
| coefficient for value loss | 1 | 1 | 1 |
| gamma | 0.99 | 0.99 | 0.99 |
| GAE lambda | 0.95 | 0.95 | 0.95 |
| use a fixed optimisation order | – | False | False |
| kl threshold | – | – | 0.01 |

Table 5: Common hyperparameters used for QMIX, QTRAN, QPLE, and WQMIX in UMAP.

|  | QMIX | QTRAN | QPLEX | WQMIX |
|---|---|---|---|---|
| optimizer | adam | adam | adam | adam |
| learning rate | 0.001 | 0.001 | 0.001 | 0.001 |
| state compat | mean observation | mean observation | mean observation | mean observation |
| hidden sizes | 128 | 128 | 128 | 128 |
| hypernet-dimension | 64 | 64 | 64 | 64 |
| TD lambda | 0.6 | 0.6 | 0.6 | 0.6 |

Table 6: Common hyperparameters used for the 15 tasks.

|  | Metal Clash | Flag Capture | Tower Challenge | Terrain Domination |
|---|---|---|---|---|
| simulation time step | 1/2560 s | 1/2560 s | 1/2560 s | 1/2560 s |
| simulation time interval | 1/2 s | 1/2 s | 1/2 s | 1/2 s |
| time dilation factor | 64 | 64 | 64 | 64 |
| parallel environment | 32 | 32 | 64 | 32 |
| maximum episode step | 125 | 250 | 100 | 150 |

Then, supplementary experiments are conducted. The mean reward for the evaluated algorithms across the first 12 tasks is plotted in Figure 12. Moreover, the result of experiments without parameter sharing is shown in Figure 13 across tasks. Generally speaking, actor-critic-based algorithms without parameter sharing perform worse than those with parameter sharing.

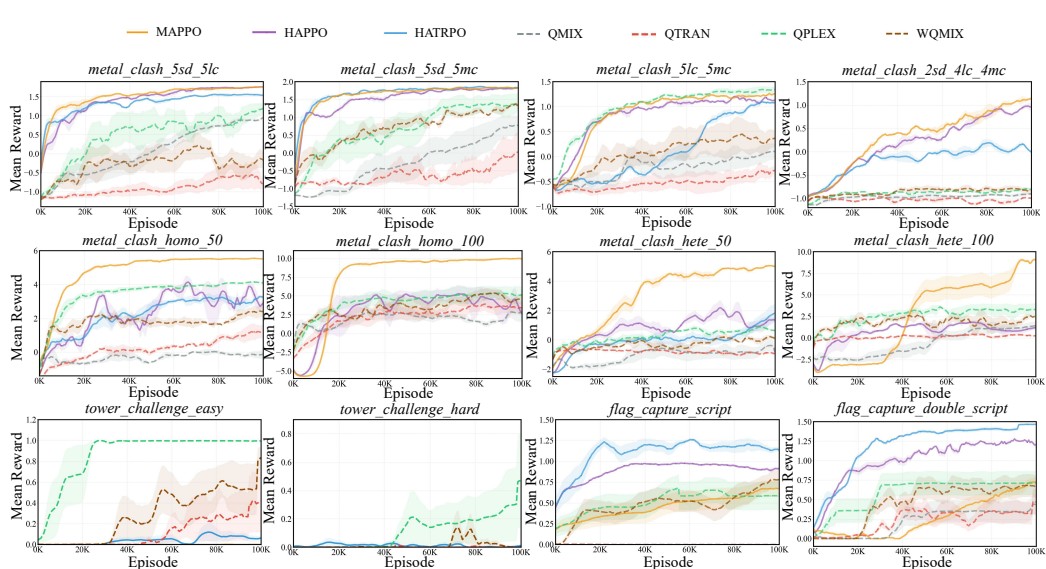

Figure 12: The comparison of reward for all evaluated algorithms across 12 tasks. The shadowed area depicts the 95% confidence interval.

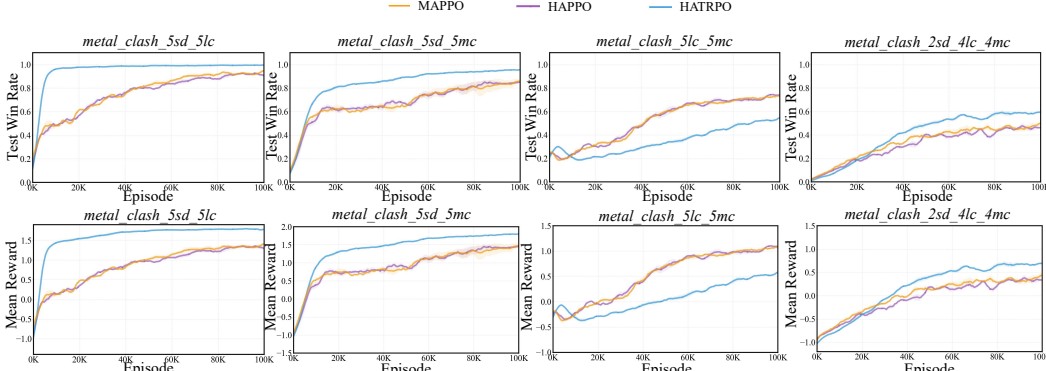

Figure 13: The comparison of test win rate and reward for actor-critic-based algorithms without parameter sharing.

