# OpenReview forum: "UMAP: A Highly Extensible and Physics-Based Simulation Environment for Multi-agent Reinforcement Learning"
_ICLR.cc/2025/Conference — Submitted to ICLR 2025_

### Official Review · Reviewer_MxdE · 2024-10-31

**Soundness:** 3
**Presentation:** 2
**Contribution:** 2
**Rating:** 3
**Confidence:** 4

**Summary:**

This paper introduces a platform for multi-agent benchmarks to be built on called UMAP. The authors argue that current MARL benchmarks are not grounded enough in real-world physics and propose leveraging Unreal Engine as a platform for building new physics-grounded tasks. They also present HMAP, a python library allowing researchers to quickly build new MARL algorithms and leverage existing ones.

**Strengths:**

Benchmarking and evaluation is an incredibly important and often underappreciated part of the ML community. Multi-agent RL in particular could benefit significantly from new and improved benchmarks that could help guide its research towards more impactful areas and to drive more fundamental/empirical advances.

Releasing a set of benchmarks alongside the experimental pipeline to easily run experiments is a big positive of this paper.

**Weaknesses:**

# Major

## References to other benchmarks
In the related work you list a lot of benchmarks but do not properly outline what is missing from some of these. For example what do you see i missing from GRF that your platform can provide?
No mention of PettingZoo, Pufferfish, JaxMARL, Robocup, or the Hanabi Challenge.

Unity ML-Agents Toolkit is also not mentioned at all in the paper, which on the surface is extremely similar (RL environments created in a gaming engine). Why are you creating a new platform using unreal engine instead of extending this one?

## Motivations
What exactly are you proposing in this paper? A new benchmark, or a platform that researchers can build their own benchmarks on? A useful benchmark that the community can utilise should be pre-specified and essentially agreed on by as many areas/researchers as possible. Being able to customise things is very helpful, but then you lose the ability to easily compare performance across the community. This paper seems to stop at providing the platform only, and seems to be expecting other researchers to build a benchmark on top. Conversely, if you are proposing a standardised benchmark then a lot more care needs to be taken over the tasks being specified, what they are testing, what evaluation procedures look like, all of which is missing in this paper. The tasks you've provided feel quite scattershot, and after reading the results I am unsure what to take away. Which area needs the most work, what are you presenting as a challenge for the community, why would increased performance in these tasks be helpful?

I would suggest being very clear on what settings you're thinking about when designing this platform. What do you mean by real-world scenarios? How many agents, what kind of tasks, etc. Be specific and provide examples. By making these concrete with a few examples, you can then contrast with other benchmarks and show what is missing to provide motivations for your platform.

Why are Points (3) and (4) advantages of UMAP? What exactly is helpful about controlling the simulation time flow, can't I record a video of any environment and watch it back slowly? Similarly, what does controllable-speed rendering mean? Are you trading off speed for visual fidelity/resource use?

## Details
Throughout the paper, there is very little mention of observation spaces. What are the inputs for the agents? Are they images or features? How do you intend or envision researchers will adjust this? Is there a particular configuration you want to see used? What about being able to access the internal state of the environment, are you allowing that? If you are proposing a benchmark, these needs to be specified in advance, and their motivations clearly outlined.

# Minor

UMAP is already a well-known method in ML. Why deliberately use it as a name for your platform?

Reference for POMGs is too new.

"...a city full of building..." - buildings?

"...a transnational airspace, or a planetary orbit..." - is the adjective transnational necessary? What do you mean by a planetary orbit? The path a planet could take? Then isn't the map much more than just the orbit?

What is different about metal clash compared to SMAC/SMACv2?

**Questions:**

Some smaller questions, there are others throughout the weaknesses I would like to see addressed particularly on missing references and motivations.

- Within a team, can different agents have different reward functions? Some more explanation on why a user would want to leverage the concept of team would be helpful.

- Can you comment on your results in Figure 4 compared to "Rethinking the Implementation Tricks and Monotonicity Constraint in Cooperative Multi-Agent Reinforcement Learning".

- What kind of hyperparameter sweep/tuning did you do for the algorithms you benchmarked?

---

> ### Author Response · Authors · 2024-11-22
> **Author Response to Reviewer MxdE (Part 1)**
>
> Thank you for your valuable feedback. Here is a detailed, **point-by-point response** addressing your main concerns, minor issues and questions.
>
> ---
>
> **Q1: Concerns about references to other benchmarks**
>
> **A1:**
>
> **a. In the related work you list a lot of benchmarks but do not properly outline what is missing from some of these. For example what do you see is missing from GRF that your platform can provide?**
>
> In the appendix of our paper, the section on related work introduces simulation environments within the MARL domain, categorized by whether incorporating physics engines. It is important to emphasize that **Table 1 in the paper has already outlined the missing parts of existing benchmarks.** Taking GRF as an example, its limited football match scenarios prevent it from simulating large-scale tasks or mixed-team game tasks. Despite being supported by a physics engine (developed by an author other than GRF's), this engine has lost some materials and is difficult to customize adequately [1]. Moreover, GRF does not support controllable simulation speed; the flow of time within its engine cannot be controlled and is subject to the clock frequency of the computing device.
>
> [1] Schuiling, B. K. 2017. GameplayFootball.github.com/BazkieBumpercar/GameplayFootball.
>
> **b. No mention of PettingZoo, Pufferfish, JaxMARL, Robocup, or the Hanabi Challenge.**
>
> We have two criteria for selecting related works for comparison. The first is that the work should be specifically designed for MARL, and the second is that the work has been peer-reviewed and accepted at top conferences or journals.
>
> Among the numerous works mentioned by the reviewers, PettingZoo and Pufferfish resemble collections of existing environments, containing many single-agent environments and overlapping with MARL environments already mentioned in the related works, such as MPE (PettingZoo), MAgent (PettingZoo & Pufferfish), SMAC (Pufferfish), and NeuralMMO (Pufferfish).
>
> To our knowledge, most versions of RoboCup consist primarily of single-agent tasks, and recent publications in multi-agent related works do not support mainstream MARL algorithms [2,3].
>
> The reviewer also mentioned JAXMARL, which was accepted by NeurIPS 2024 this year, coinciding with the submission deadline for this conference. We will include JAXMARL and the Hanabi Challenge in our discussion of related work.
>
> [2] Reis L P. Coordination and Machine Learning in Multi-Robot Systems: Applications in Robotic Soccer[J]. arXiv preprint arXiv:2312.16273, 2023.
>
> [3] Smit A, Engelbrecht H A, Brink W, et al. Scaling multi-agent reinforcement learning to full 11 versus 11 simulated robotic football[J]. Autonomous Agents and Multi-Agent Systems, 2023, 37(1): 20.
>
> **c. Unity ML-Agents Toolkit is also not mentioned at all in the paper, which on the surface is extremely similar (RL environments created in a gaming engine). Why are you creating a new platform using unreal engine instead of extending this one?**
>
> We did not mention Unity ML-Agents in our paper because it is not specifically designed for MARL environments. The reasons for creating a new platform using Unreal Engine instead of extending the platform based on the Unity Engine are as follows:
>
> 1. **Compared to Unity Engine, Unreal Engine features a fully open-source underlying framework.** The source code of UE is easily accessible to everyone. This allows us to identify and resolve any low-level framework simulation or efficiency issues by delving into the UE source code. This is the foundation of UMAP's high extensibility.
>
> 2. **Compared to Unity Engine, Unreal Engine offers more realistic 3D graphics rendering and physical simulations.** In the game development community, Unity Engine is often used for developing cartoon-style 2D game environments (as seen in the built-in tasks of Unity ML-Agents), whereas Unreal Engine is preferred for developing realistic-style 3D games (famous examples include *Pro Evolution Soccer* and *Halo*). The physics engine in Unreal Engine can accurately simulate real-world physical laws, such as gravity, collision, and friction, which is particularly important for training tasks that require precise physical simulations. The high-quality graphic rendering capabilities provided by Unreal Engine can produce visuals close to the real world, facilitating intuitive and detailed analysis during development and use. This is the foundation for UMAP's high realism.
>
> 3. **Convenient Blueprint Programming Language**: Although not the most critical advantage, the convenient Blueprint programming language in Unreal Engine is very helpful for development. With the unique visual scripting language, Blueprint, most procedures in multi-agent environment design no longer require writing any codes. Blueprint offers drag-and-drop programming that eases the effort of researchers. In sophisticated projects, this tool significantly improves team cooperation.

---

> ### Author Response · Authors · 2024-11-22
> **Author Response to Reviewer MxdE (Part 2)**
>
> **Q2: What is the motivation behind this article? Is it a platform that can create environments, or is it a collection of tasks that are already usable?
> The provided built-in tasks seem quite scattershot, and after reading the results I am unsure what to take away.**
>
> **A2:**
> This is a meaningful question. Clarifying the contribution of a work and its unique position within the entire research community is very important. UMAP is specifically designed for MARL, with its irreplaceability reflected in the high extensibility while based on a 3D physical engine. According to the classification in Unity ML-Agents, it is more akin to a general platform. The term "platform" was not used in our paper due to concerns about potential confusion with the concept of "cross-platform rendering."
>
> However, please note that the built-in scenarios and tasks of UMAP are not "quite scattershot." The design principle is that, since UMAP is an extensible environment specifically designed for MARL, its built-in tasks  should highlight the unique challenges of MARL compared to RL, rather than problems already existing in the RL domain. These challenges arise due to transitions：
>
> -	from single agent to multiple agents (*scale perspective*),
> -	from homogeneous agents to heterogeneous agents (*heterogeneity perspective*),
> -	from individual reward to group rewards (*credit assignment perspective*),
> -	from single objective to mixed objectives (*multi-team perspective*).
>
> Based on the experimental results of the tasks mentioned above, no current SOTA algorithm can achieve best performance across all tasks. Therefore, there is a need to design new tasks and test new algorithms in highly extensible environments, which from another perspective proves the importance of UMAP.
>
> Thank you for the detailed suggestion. We will modify the descriptions in the introduction and related work to make the overall motivation of UMAP more concrete.

---

> ### Author Response · Authors · 2024-11-22
> **Author Response to Reviewer MxdE (Part 3)**
>
> **Q3: Why are Points (3) and (4) advantages of UMAP? What exactly is helpful about controlling the simulation time flow, can't I record a video of any environment and watch it back slowly? Similarly, what does controllable-speed rendering mean? Are you trading off speed for visual fidelity/resource use?**
>
> Firstly, controllable time flow is a unique and highly useful feature of UMAP. The main functions of this feature are outlined below:
>
> - **Accelerating Simulation Speed (Training)**: UMAP controls the time flow of simulations through a time dilation factor. Time dilation factor provides another dimension to speed up training besides increasing the number of parallel processes. With the same number of parallel processes, increasing the time dilation factor can make individual processes faster, thereby enhancing overall training efficiency.
>
> - **Optimal Use of Computing Resources (Training)**: With the same number of parallel processes, changing the time dilation factor does not significantly affect the use of memory or GPU memory, but it does affect CPU utility (details can be found in our response to reviewer XX). This is very user-friendly for researchers with limited resources. When memory resources are constrained, increasing the time dilation factor allows for full utilization of CPU resources; when CPU resources are limited, reducing the time dilation factor and increasing the number of parallel processes can "trade memory for time", thus avoiding the waste of other computing resources when some reach their limits.
>
> - **Precise Time Matching (Execution)**: UMAP's highly optimized time control system allows for precise matching of simulation time with real time, which helps in reducing the sim-to-real gap.
>
> - **Slow Motion Analysis (Execution)**: Combined with the rendering mechanism, low-speed time flow simulation aids in-depth analysis of the environment, providing timely feedback for algorithm debugging and environment development.
>
> Secondly, the rendering mechanisms of UMAP greatly facilitate environment development and algorithm debugging. UMAP has two basic rendering approaches:
>
> 1. **Rendering within Unreal Editor**: This allows developers to directly invoke algorithms and policy models within HMAP in the editor, and control the editor's assets directly using Python. This eliminates the need for manual test case setups, significantly enhancing development convenience.
>
> 2. **Rendering with Compiled Binary Files**: Users can directly invoke and load policy models in HMAP, and use an executable file to render HMAP data in real-time (just by copying and executing a single *bash command*). With this approach, users can observe the on-going training process in real-time, even conducting multi-threaded training on a high-performance server and then rendering the training status of a particular thread on a local computer in real-time.
>
>
> Additionally, integrating rendering with controllable time flow significantly facilitates debugging environments and algorithms. **This approach is fundamentally different from the "record video and then play back in slow speed" method mentioned by the reviewer.** Regardless of the rendering mode chosen, users can freely change the camera angle and zoom in or out. The Unreal Engine can automatically adjust rendering details based on the scale. What's more, by changing the time dilation factor, one can slow down to observe interactions between specific agents or speed up to view the macro situation of the entire multi-agent system.

---

> ### Author Response · Authors · 2024-11-22
> **Author Response to Reviewer MxdE (Part 4)**
>
> **Q4:What is the observation space of an agent? How do researchers adjust the observation space? Is there a particular configuration you want to see used? Does UMAP support accessing the internal state of the environment?**
>
> 1. To adapt to the current mainstream MARL algorithm frameworks, UMAP's existing built-in scenarios all use a vector-form of observation structure. However, we have also reserved a general interface on the agent perception module to support more diverse forms of observations, such as text and potentially images.
>
> 2. In most cases, researchers can fully modify the observation space and observation functions of agents through the Python interface layer. However, more in-depth observation logic needs to be modified through the advanced module layer. For example, an agent might be able to observe another agent through one layer of glass but not through two layers. This requires modifications to the agent observation module to achieve.
>
> 3. We are not completely sure what "particular configuration" mean. If it refers to fixed/similar structure observations, that is not what we aim to utilize, as it does not benefit UMAP's extensibility. However, users have fixed configurations available when constructing observation spaces and functions, including map-level information, individual agent information, and special object information. Users can utilize the aforementioned information to build rich observation functions and extensions.
>
> 4. UMAP supports accessing the internal state of the environment. The internal information of the environment can be transmitted from the UE side to the Python side through a fixed map-level interface or by setting special object information.

---

> ### Author Response · Authors · 2024-11-22
> **Author Response to Reviewer MxdE (Part 5)**
>
> **Q5：Minor Concerns**
>
> **A5:**
>
> **a. UMAP is already a well-known method in ML. Why deliberately use it as a name for your platform?**
>
> We have chosen the names *UMAP* and *HMAP* for their brevity and recognizability. However, it was pointed out that UMAP is previously known as a method for data dimension reduction, whereas ours is an environment within the MARL domain, which should be distinct enough to avoid confusion under normal circumstances. We appreciate the reviewer's comment on this matter and will consider renaming them after the review process to prevent any potential misunderstandings.
>
> **b. Reference for POMGs is too new.**
>
> The reason for citing more recent literature is that this work incorporates elements of partial observability that may arise in multi-agent systems and utilizes agent-level notation, thereby enhancing its universality. We will add the original citation of the POMG literature [1] on top of the existing references.
>
>
> **c. "...a city full of building..." - buildings?
> "...a transnational airspace, or a planetary orbit..." - is the adjective transnational necessary? What do you mean by a planetary orbit? The path a planet could take? Then isn't the map much more than just the orbit?**
>
> We appreciate the feedback on the inappropriate terminology used in the text. The examples of "airspace" and "planetary orbit" were intended to emphasize the vast potential applications of the map. Indeed, the adjective "transnational" is not suitable in this context. We will remove some statements unrelated to the main task of this paper to avoid such confusion.
>
> **d. What is different about metal clash compared to SMAC/SMACv2?**
>
> This is an interesting question. *Metal-clash* is our first scenario developed using UMAP, inspired indeed by the SMAC based on StarCraft. However, unlike SMAC, *metal-clash* specifically emphasizes the challenges of heterogeneity and large-scale in MARL. Regarding heterogeneity, each type of basic agent in *metal-clash* has significantly different observation spaces and functions, manifested in 1. different observation ranges, 2. different maximum numbers of entities that can be recorded, and 3. different recording precision for each entity. This differs from the classic environments like SMAC, where heterogeneous agents share observation spaces and functions [2]. In terms of the challenge of large-scale, *metal-clash* supports the interaction of hundreds of agents in the scenario, whereas the official maps of SMAC/SMACv2 do not have more than 100 agents interacting.
>
> Moreover, thanks to the high scalability of UMAP, *metal-clash* can easily modify the number, type, and attributes of agents on the Python side, while SMAC requires a map editor to change the number and type of agents in the mission, and it does not allow for deep customization of agents (game units).
>
> [1] Littman M L. Markov games as a framework for multi-agent reinforcement learning[M]//Machine learning proceedings 1994. Morgan Kaufmann, 1994: 157-163.
>
> [2] Yu X, Lin Y, Wang X, et al. GHQ: Grouped Hybrid Q Learning for Heterogeneous Cooperative Multi-agent Reinforcement Learning[J]. arXiv preprint arXiv:2303.01070, 2023.

---

> ### Author Response · Authors · 2024-11-22
> **Author Response to Reviewer MxdE (Part 6)**
>
> **Q6: Within a team, can different agents have different reward functions? Some more explanation on why a user would want to leverage the concept of team would be helpful.**
>
> **A6:**
> Yes, within the same team, agents are allowed to have different reward functions. The training objective for each team is to maximize the sum of the cumulative average returns of the entire team. As described in Section 2, the reward function is at the agent level rather than at the team level or global level. This modeling approach is more universally applicable.
>
> The concept of a team stems from the extension of the game among multiple agents to the game among multiple groups. The game among multiple agents can be modeled through the original Markov game, but when agents become groups and the influence of individuals within a group cannot be ignored, the game among multiple groups requires the introduction of the concept of a team to assist in modeling. For example, in a stock market, the game among individual investors can be modeled through the game among multiple agents. However, when considering investment groups composed of multiple individuals, the game among investment groups cannot be modeled using the conventional multi-agent model, as investment groups cannot be regarded as individual agents like individual investors. Introducing the concept of a team can effectively model the agents within an investment group and the entire group.
>
> ---
>
> **Q7: Can you comment on your results in Figure 4 compared to "Rethinking the Implementation Tricks and Monotonicity Constraint in Cooperative Multi-Agent Reinforcement Learning"?**
>
> **A7:**
> The paper mentioned by the reviewer delves deeply into the work on value decomposition within the MARL domain. It discusses the relationship between different algorithms (such as QMIX, QPLEX, VDN) in terms of the strength of value decomposition constraints and their actual performance. It concludes that in cooperative scenarios, as the monotonicity constraint decreases, performance worsens. Our experiments also employed four value decomposition algorithms: QMIX, QPLEX, QTRAN, and W-QMIX. Based on our results, our findings seem to be inconsistent with theirs. However, it is important to emphasize that in their experiments, due to the simplicity of the tasks, the actual differences between different algorithms were very small, with overall performance fluctuating roughly between 91% and 95%. In our task experiments, these algorithms did not achieve such high win rates universally. Their findings might be patterns specific to certain scenarios, which is a domain worthy of further investigation.
>
> ---
>
> **Q8: What kind of hyperparameter sweep/tuning did you do for the algorithms you benchmarked?**
>
> **A8:**
> We employed a grid search method for hyperparameter tuning. In fact, in the experimental section, to ensure fairness and rigor in comparison, we did not use our built-in MARL algorithms of HMAP for comparison. Instead, we utilized algorithms from third-party frameworks. The vast majority of the hyperparameters we used in our experiments were kept consistent with the original hyperparameters within these frameworks.
>
> ---
>
> Hope the above responses have clarified our work and addressed your concerns. We **welcome any further questions you might have** and thank you for your dedication!

---

> ### Author Response · Authors · 2024-11-29
> **Request for Response**
>
> Dear Reviewer MxdE,
>
> We sincerely request your attention to review our responses to the concerns you have raised. We are eager to learn whether our response has addressed your concerns and if there are any further questions you might have.
>
> Moreover, we have made revisions to the manuscript before the deadline as promised. The major changes related to your concerns include:
>
> 1. The addition of referenced related work and a detailed discussion in the appendix on aspects missing in each work compared to UMAP.
>
> 2. An expanded introduction to each built-in scenario in the appendix, including explanations of observations, actions, and rewards.
>
> 3. The inclusion of experiments on UMAP's training efficiency and resource consumption in the appendix, demonstrating the advantages of UMAP's controllable time flow for training.
>
> 4. Modifications to address minor issues you mentioned, such as references and unclear examples.
>
> We look forward to your valuable feedback.

---

> > ### Comment · Reviewer_MxdE · 2024-11-29
> > **Reply**
> >
> > Thanks for your reply.
> >
> > 1. "We did not mention Unity ML-Agents in our paper because it is not specifically designed for MARL environments." - This is not enough of a reason to completely ignore any mention of Unity ML-Agents in this paper. Unity ML-Agents can be used to create a multi-agent environment (https://unity-technologies.github.io/ml-agents/ML-Agents-Overview/#training-in-cooperative-multi-agent-environments-with-ma-poca as one example).
> > 2. I disagree that Unity cannot be used to develop games with non-cartoon graphics or decent physics simulations. Unity features a physics engine, and many games have been made that are not "cartoon-style 2D game environments" (Escape from Tarkov as a single example).
> > 3. If you are positioning UMAP as a new multi-agent benchmark, then (as mentioned by other reviewers) more focus needs to be on the evaluation and reasoning for the tasks chosen. However, in the replies it still seems that UMAP is more of a platform designed for other researchers to build environments on top off.
> > 4. "we did not use our built-in MARL algorithms of HMAP for comparison." - If HMAP is part of the your contributions, then shouldn't you benchmark with it as well?

---

> > > ### Author Response · Authors · 2024-11-30
> > > **Author Response to Reviewer MxdE (Part 8)**
> > >
> > > **Q11: If you are positioning UMAP as a new multi-agent benchmark, then (as mentioned by other reviewers) more focus needs to be on the evaluation and reasoning for the tasks chosen. However, in the replies it still seems that UMAP is more of a platform designed for other researchers to build environments on top off.**
> > >
> > > **A11:**
> > > Thank you for your suggestion. As mentioned in our rebuttal, we position UMAP as a general platform specifically designed for MARL, rather than a new multi-agent benchmark. The development of learnable built-in scenarios (or example environments) and related tasks is to demonstrate UMAP's potential for developing MARL tasks, including features such as heterogeneous, large-scale, or multi-team.
> > >
> > > We describe UMAP as a highly extensible simulation environment in the paper because
> > > users can not only develop completely new scenarios based on UMAP, but also deeply modify and even combine existing built-in scenarios. The high extensibility of the built-in scenarios is reflected at three levels:
> > > 1. (Parameter-level) Users can directly modify the number of agents, team affiliations, and basic properties of each agent within each scenario through hyperparameters.
> > > 2. (Function-level) Users can directly modify Python code to change the observation functions, reward functions, and agent initialization functions corresponding to each scenario.
> > > 3. (Task-level) Users can directly transplant the map or basic agents of one scenario to another for use (details in Appendix C.1).
> > >
> > > All the above operations for extending built-in scenarios can be implemented solely through Python, a capability not possessed by general platforms like Unity ML-Agents. These built-in scenarios, along with other structures of UMAP, form a highly extensible environment. Thank you once again for your suggestion, we will consider modifying the relevant statements in the paper.
> > >
> > > **Q12: "we did not use our built-in MARL algorithms of HMAP for comparison." - If HMAP is part of the your contributions, then shouldn't you benchmark with it as well?**
> > >
> > >
> > > **A12:**
> > > It may be that our expression in the rebuttal led you to misunderstand our intention. During the early development phase of HMAP, we implemented MAPPO, HAPPO, and other MARL algorithms for functional debugging. However, it would be unfair to compare using our own unofficial algorithms, so we added wrappers to HMAP for compatibility with third-party frameworks, allowing algorithms from popular open-source code libraries to be directly deployed for training based on HMAP. Now, these algorithms from third-party frameworks have covered HMAP's built-in MARL algorithms.
> > >
> > > ---
> > >
> > > Thank you again for your valuable questions and constructive comments. If you have any other unresolved concerns and questions, we would be very happy to reply to you before the discussion phase ends.

---

> > > ### Author Response · Authors · 2024-12-03
> > > **Request for Response**
> > >
> > > Dear Reviewer MxdE,
> > >
> > > We would like to inquire whether our previous responses have adequately addressed your concerns. If you have any additional questions or concerns, we kindly request that you share them with us before the end of the discussion phase, so we can take advantage of the final day to provide our responses.
> > >
> > > We greatly appreciate your valuable feedback and look forward to your further comments.

---

> ### Author Response · Authors · 2024-11-30
> **Author Response to Reviewer MxdE (Part 7)**
>
> Thanks for your reply. We hope our previous response has already addressed some of your concerns. We will now respond to the four issues you mentioned.
>
> ---
>
> **Q9: Neglect of mention regarding Unity ML-Agents in the paper.**
>
> **A9:** We appreciate you pointing this out. We will explicitly add a discussion on Unity ML-Agents in the related work part. Unity ML-Agents is an RL-related work with a design philosophy very similar to that of UMAP.
>
> We will clearly highlight the distinct advantages of UMAP (which developed based on the Unreal Engine), compared to Unity ML-Agents (which developed based on Unity):
>
> 1. The underlying code of the Unreal Engine is completely open-source. With such fully open-source UE source code, we can directly modify the underlying C++ framework to address simulation-related and efficiency-related issues. This ensures the layered architecture design of UMAP, allowing users to focus only on the *Python interface layer* and the *advanced module layer*. Similarly, we have addressed the challenges arising from increasing the number of agents, specifically the dramatic increase in computational load of the low-level perception detection module and the significant overhead in inter-process communication (for more details, please refer to our response to Reviewer tZCz Q3&A3). These improvements enable UMAP to effectively simulate scenarios with a larger number of agents.
>
> 2. Convenient Blueprint programming language. As a graphical programming method, the language within the Unreal Editor not only significantly reduces the workload of us in developing UMAP, but also facilitates in-depth customization by users. Unlike Unity ML-Agents, users do not need to use C# or other codes when developing a new scenario with UMAP; instead, they only need graphical programming, which greatly lowers the barrier to use.
>
> 3. Comprehensive optimization for MARL. Beyond the aforementioned low-level optimizations for multi-agent system simulation, both UMAP's architectural design and its integration with HMAP are specifically tailored to serve the MARL research community. Our work allows UMAP to be directly integrated with the vast majority of popular MARL algorithms, which is a significant advantage not possessed by Unity ML-Agents.
>
> **Q10: I disagree that Unity cannot be used to develop games with non-cartoon graphics or decent physics simulations. Unity features a physics engine, and many games have been made that are not "cartoon-style 2D game environments" (Escape from Tarkov as a single example).**
> **A10:** We did not claim that Unity cannot be used for developing scenes beyond cartoon-style 2D games. Hence, we used words like "often" and "preferred" in our discussion to reflect a community perspective. However, notable works such as AirSim[1] have pointed out their reasons for choosing the Unreal Engine over other game engines (including Unity), which include "being an open source and available on Linux, Windows as well as OSX", "bringing some of the cutting-edge graphics features", and "near photo-realistic rendering capabilities." Considering whether a game engine has higher graphic rendering quality and physical realism is unrelated to the issues addressed in our work, we will ensure that similar discussions do not appear in our papers.
>
> [1] Shah S, Dey D, Lovett C, et al. Airsim: High-fidelity visual and physical simulation for autonomous vehicles[C]//Field and Service Robotics: Results of the 11th International Conference. Springer International Publishing, 2018: 621-635.

---

### Official Review · Reviewer_tZCz · 2024-11-01

**Soundness:** 4
**Presentation:** 3
**Contribution:** 3
**Rating:** 8
**Confidence:** 3

**Summary:**

This paper proposes UMAP which is an extensible, physics-based 3D simulation environment implemented on the Unreal Engine. UMAP offers (1) diverse multi-agent tasks, (2) customizable task design, (3) controllable simulation speeds, and (4) rich, cross-platform rendering capabilities. The authors compare UMAP with existing environments, emphasizing its flexibility and extensibility.

**Strengths:**

- The authors conduct physical experiment to demonstrate the potential of UMAP in bridging the sim-to-real gap. The integration of a motion capture system and communication between UMAP and real-world environments provides a framework validates the efficacy of the proposed system in replicating multi-agent policies from a virtual to a physical setup.
- Built on Unreal Engine, UMAP is highly extensible, providing a flexible foundation for further development and integration with HMAP.
- Based on UMAP, the authors design a series of foundation tasks including heterogeneity, large scale, sparse team rewards, and multi-team. These tasks impose higher requirements on cooperation and competition among multi agents compared to existing tasks.

**Weaknesses:**

- This paper does not explain how the physics simulation in Unreal Engine contributes to the sim-to-real gap or why Unreal Engine was chosen as the simulation software.
- The extensibility comes at the cost of increased complexity, which could hinder adoption among non-expert users or practitioners without a strong technical background.

**Questions:**

- What specific limitations are associated with UMAP when handling large-scale tasks (e.g., with agents over 100)?
- How would the framework perform in dynamic settings where data distributions change over time? Is UMAP adaptable to such scenarios?

---

> ### Author Response · Authors · 2024-11-22
> **Author Response to Reviewer tZCz (Part 1)**
>
> Thank you for your valuable feedback. Here is a detailed, **point-by-point response** addressing your main concerns, minor issues and questions.
>
> ---
>
> **Q1：This paper does not explain how the physics simulation in Unreal Engine contributes to the sim-to-real gap or why Unreal Engine was chosen as the simulation software**
>
> **A1:**
> The choice of Unreal Engine (UE) brings many benefits, not only aiding in narrowing the sim-to-real gap but also enhancing scalability and the creation of rich tasks. Here we list some typical advantages:
>
> 1. **Accurate built-in physical simulation**: The physics engine in UE can accurately simulate real-world physical laws, including gravity, collision, friction, etc., which is particularly important for training tasks that require precise physical simulations.
> 2. **Rich resources and plugins**: The Unreal community boasts a wealth of resources, including 3D models, animations, and accessible solutions, which are crucial for developing environments with scalability.
> 3. **Highly realistic graphic rendering**: The high-quality graphic rendering capabilities provided by UE can produce visuals close to the real world. This facilitates intuitive and detailed analysis during development and use, such as slow-motion analysis to observe differences between the current virtual system and the real system.
>
> Furthermore, compared to other game engines or physics engines like Unity3D, UE also has its unique advantages, further reinforcing our reason for choosing it as our simulation software.
> 1. **Fully open-source underlying framework**: The source code of UE is easily accessible to everyone. This means that we can identify and resolve any low-level framework simulation/efficiency issues by diving into the UE source code. In fact, we have utilized this open-source mechanism to complete the first and second layers of UMAP (details of modifications to the original UE code can be found in Appendix C.1).
> 2. **Convenient Blueprint programming language**: With UE's unique visual scripting language, Blueprint, most procedures in multi-agent environment design no longer require writing any codes. UE Blueprint offers drag-and-drop programming that eases the effort of researchers. In sophisticated projects, this tool significantly improves team cooperation.
>
> ---
>
> **Q2: The extensibility comes at the cost of increased complexity, which could hinder adoption among non-expert users or practitioners without a strong technical background.**
>
> **A2:**
> It is important to emphasize that while the scalability of UMAP indeed introduces a significant increase in complexity for the developers (building all five layers of UMAP and its ecosystem), **it does not significantly increase the difficulty for users.**
> We have made three main efforts to make UMAP more accessible to non-expert users.
>
> -	**Firstly, we adopted a hierarchical design of UMAP.** We have made great efforts to ensure the universality of the Python interface layer, allowing users to cover modifications of all elements within a POMG directly through this layer. With built-in maps and agents, users can easily develop tasks similar to MPE or SMAC using the top-level Python layer of UMAP.
>
> -	**Secondly, we developed a framework fully compatible with UMAP, named HMAP.** With HMAP, users can easily modify task elements, deploy algorithms, and fully utilize UMAP's rich rendering mechanisms with minimal effort. This framework is also compatible with third-party environments and algorithms from other frameworks, ensuring that users' existing research workflows are not disrupted.
>
> -	**Thirdly, we developed a related ecosystem and comprehensive tutorials.** We have created Docker images that include the entirety of HMAP, the compiled files and Python layer of UMAP, allowing users to deploy quickly without worrying about the workload of environment configuration. Moreover, we have prepared comprehensive tutorials to facilitate users, including tutorials on using, debugging, developing new tasks on UMAP, as well as using and debugging HMAP. These tutorials, including PPTs and videos, will be made public after the review process.

---

> ### Author Response · Authors · 2024-11-22
> **Author Response to Reviewer tZCz (Part 2)**
>
> **Q3：What specific limitations are associated with UMAP when handling large-scale tasks (e.g., with agents over 100)?**
>
> **A3:**
> The computation within each agent is automatically managed in parallel by the Unreal Engine, thus this part does not contribute to the limitations of large-scale simulations.
> During our testing phase, we have identified two primary bottlenecks:
>
> 1. **Agent-wise Perception Modules**: The challenge arises when managing a large number of agents (e.g., 200 agents), and each agent with potentially different perception distances and shapes (e.g., cone, sphere, etc.). It necessitates the computation of a 200x200 perception matrix. To address this issue, we have implemented several strategies, including:
>     - Batched distance computation utilizing the xtensor library.
>     - Parallel processing of perception computations.
>     - Redesign of the existing Unreal Engine built-in agent perception pipeline to optimize performance.
>
> 2. **Inter-Process Communication (IPC) Time Consumption**: IPC between Python and the UE simulation becomes a bottleneck as the scale of the simulation increases. While IPC costs can be ignored for smaller team sizes due to efficient handling by the operating system, they become significant when the number of agents exceeds 100, leading to a substantial increase in IPC package size. Our current strategies for mitigating this issue include:
>     - Utilization of fast compression algorithms, such as lz4, to reduce data size.
>     - Development of shared memory implementations to further enhance efficiency and reduce IPC time consumption.
>
> ---
>
> **Q4：How would the framework perform in dynamic settings where data distributions change over time? Is UMAP adaptable to such scenarios?**
>
> **A4:**
> Yes, UMAP support dynamic settings in two levels.
>
> 1. **Distribution Shift Over Episodes**: Agents' attributes, such as size, perception range, and spawn locations, can be conveniently modified at the start of a new episode from the Python side. This process does not require re-compiling the UE simulation client, making it highly efficient for iterative testing and development.
>
> 2. **Distribution Shift Within an Episode**: Achieving distribution shifts within an episode is also straightforward. By leveraging the *event system*, special events can be added on the UE side to facilitate timestep-level data distribution shifts. These operations can be efficiently executed through UMAP's advanced module layer. For more precise control over distribution, a new set of special actions can be designed. This allows for intricate distribution control directly from the Python side, enhancing the flexibility and depth of simulation customization.
>
> ---
>
> Hope the above responses have clarified our work and addressed your concerns. We **welcome any further questions you might have** and thank you for your dedication!

---

### Official Review · Reviewer_1ovJ · 2024-11-01

**Soundness:** 3
**Presentation:** 2
**Contribution:** 3
**Rating:** 6
**Confidence:** 3

**Summary:**

This paper introduces a new simulation environment, Unreal Multi-Agent Playground (UMAP), for multi-agent reinforcement learning, which is designed to overcome limitations in authenticity and complexity present in existing platforms.
UMAP offers a suit of physics-based 3D environments that involve heterogeneous agents, large-scale simulations, and multiple teams. UMAP consists of a hierarchical five-layer architecture, while the interface layer is exposed to end users to easily customize tasks using Python. Authors also release the MARL experiment framework, HMAP, to facilitate the use of UMAP. Extensive experiments are done on different tasks in UMAP, and results reveal that there is still the need for more advanced algorithms to dominate those tasks. In addition, authors construct a digital-twin of a real-world task in UMAP to demonstrate its sim-to-real ability.

**Strengths:**

1. The paper does a nice job of presenting its ideas clearly and making the technical aspects easy to follow.

2. UMAP tackles an important issue in the field: the struggle to make MARL simulations relevant for real-world applications. By creating a suit of realistic and flexible environments, UMAP makes substantial contribution to pushing MARL closer to real-world tasks.

3. The UMAP framework is solid, and diverse built-in scenarios gives researchers options for testing algorithms under different settings, making UMAP a valuable tool.

**Weaknesses:**

1. The authors did not benchmark the efficiency of the UMAP simulation. Considering that many recent simulation frameworks (e.g., IsaacGym, IsaacLab, SAPIEN) support GPU parallelization, I am curious whether UMAP could potentially offer similar support. Such high-efficiency simulation has been shown to be highly beneficial for training RL policies in single-agent tasks.

2. The experiments on the physical environment in Section 6.4 do not support the authors claim on sim-to-real ability. The authors merely present rendered images from the sim and real environments without providing any results related to policy learning. For instance, they do not show how a policy trained in simulation performs in the real environment or whether there is a strong correlation between the performance of different policies in simulation and their performance in the real environment.

**Questions:**

1. Considering the recent studies on offline MARL, do the authors have any plans to release offline datasets based on the UMAP environment and evaluate the performance of offline algorithms?

2. What would be the workload involved in designing a new environment? For example, if custom maps or objects need to be imported, and their properties set, what steps are required? Specifically, which parts must be modified through the Unreal Engine, and which can be directly configured through the Python interface?

If the authors address the above concerns, I am happy to raise the score.

---

> ### Author Response · Authors · 2024-11-22
> **Author Response to Reviewer 1ovJ (Part 1)**
>
> Thank you for your valuable feedback. Here is a detailed, **point-by-point response** addressing your main concerns, minor issues and questions.
>
> ---
>
> **Q1: What about the efficiency of the UMAP simulation? Does UMAP potentially offer similar support like GPU parallelization?**
>
> **A1:**
> This question is of great value. Overall, as a simulation environment based on a 3D physical engine, **UMAP boasts high simulation efficiency. It is well-designed to adapt to and fully utilize various types of computing resources.** Although UMAP does not support full GPU-based parallelization, it supports multi-threaded parallel simulation. Additionally, with the feature of time dilation factors, UMAP can not only improve simulation efficiency by increasing the number of parallel processes, but also **control the simulation speed of each process to make full use of computing resources (using more CPU utilization under the same memory and GPU memory), which is a functionality not available in other simulation platforms.**
>
> To verify the efficiency of UMAP, we conducted experiments on a Linux server equipped with AMD7742 CPU (maximum frequency 2.25GHz) and NVIDIA RTX3090 GPUs. The experiments, by setting different numbers of parallel processes and time dilation factors, assessed the simulation efficiency (FPS) and resource consumption indices. The QMIX algorithm from the PyMARL2 framework was tested on the *metal_clash_5sd_5mc* task. To ensure the rigor of the experiments, we made the following settings:
> 1. Except for the number of parallel processes and time dilation factors, the hyperparameters on the algorithm side and the environment side were kept consistent.
> 2. Experiments regarding each set of parallel process numbers and time dilation factors were conducted 5 times to take the average value.
> 3. Before each experiment began, the server was kept idle, running only basic system processes.
>
> Below are the results of changing the time dilation factor while keeping the number of parallel processes constant:
>
> **PyMARL2-QMIX-8-Parallel-Envs**
>
> | Time Dilation Factor | 0.5 | 1.0 | 2.0 | 4.0 | 8.0 | 16.0 | 32.0 | 64.0 |
> |----------------------|-----|-----|-----|-----|-----|------|------|------|
> | FPS (virtual timestep per real second) | 10.17 | 22.78 | 40.82 | 82.33 | 158.76 | 285.31 | 384.19 | 643.43 |
> | CPU Utility (%) | 0.1 | 0.2 | 0.4 | 0.6 | 1.1 | 1.6 | 1.7 | 1.9 |
> | Memory (MB) | 4826 | 5034 | 5158 | 5176 | 5225 | 5312 | 5358 | 5585 |
> | GPU Memory (MB) | 1531 | 1531 | 1531 | 2061 | 2139 | 2161 | 2161 | 2161 |
>
> Below are the results of changing the number of parallel processes while keeping the time dilation factor constant:
>
> **PyMARL2-QMIX-32-Time-Dilation-Factor**
>
> | Parallel Environments | 8 | 16 | 32 | 64 | 128 |
> |-----------------------|---|----|----|----|-----|
> | FPS (virtual timestep per real second) | 384.19 | 528.57 | 696.35 | 850.43 | 1044.46 |
> | CPU Utility (%) | 1.7 | 2.2 | 3.3 | 5.3 | 9.2 |
> | Memory (MB) | 5358 | 7774 | 9425 | 13285 | 18782 |
> | GPU Memory (MB) | 2161 | 2413 | 2767 | 3525 | 3703 |
>
> From the above results, we can draw the following conclusions:
> 1. By increasing the number of parallel processes and the time dilation factor, training efficiency can be significantly improved.
> 2. Increasing the time dilation factor significantly affects the CPU usage and FPS, without significantly increasing memory and GPU memory usage. Before reaching the limit of CPU single-core resources (limited by the CPU single-core maximum frequency), the CPU usage and FPS are roughly proportional to the time dilation factor.
>
> Conclusion 2 means that under limited memory resources, training efficiency can be improved by increasing the time dilation factor to fully utilize CPU resources; similarly, under limited CPU computing resources, reducing the time dilation factor and increasing the number of processes can avoid the waste of computing resources.
>
> In fact, when the number of processes is 8 and the time dilation factor is 32, training 1024 episodes on the *metal_clash_5sd_5mc* task takes less than 2 minutes. **This means that under such parameter settings, this server can simultaneously support 50 such tasks (each with 20 agents) and complete all training tasks (100k episodes) within 3 hours.** In special cases, the number of parallel processes can be further increased to improve training efficiency. When the number of processes reaches 128 and the time dilation factor is set to 32, the FPS can reach 1000+, and the training task can be completed in about an hour.
>
> Please note that the FPS here counts the number of virtual UMAP timesteps per real second. Considering this is a simulation of 20 agents, and **each timestep in UMAP undergoes 1280 frames of calculations for environmental dynamics and kinematics to maintain fine state transitions** (details in Appendix C.2), this is already highly efficient computation.

---

> ### Author Response · Authors · 2024-11-22
> **Author Response to Reviewer 1ovJ (Part 2)**
>
> **Q2: The experiments on the physical environment in Section 6.4 do not support the authors claim on sim-to-real ability.**
>
> **A2:**
> Our core argument regarding UMAP's sim-to-real ability encompasses two main points. Firstly, UMAP can serve as a bridge for deploying virtual policies into real-world scenarios. Secondly, UMAP facilitates the creation of a virtual environment that replicates real-world settings, where policy trained in this virtual environment remain effective in real scenarios.
>
> In the experimental section, we aim to demonstrate both points by constructing a virtual environment for training, followed by deploying the trained policy into real-world scenarios and showing its effectiveness. It is important to highlight that the top row of subfigures in Figure 5 displays the rendering results of policy executed solely in the virtual environment (performance in the virtual setting), while the bottom row showcases the results of deploying policy in real-world scenarios at the corresponding timesteps (performance in the real-world setting). Although there is also a virtual environment for computation and communication with HMAP during the deployment of policy in real scenarios, the information of this virtual environment is synchronized with that of the real environment at every timestep. This virtual environment is not shown in Figure 5. Therefore, the high consistency in the trajectories of agents at the corresponding timesteps demonstrates the strong correlation between the performance of policy in virtual settings and their effectiveness in real-world settings.
>
> What’s more, to further address your concern, we tested the winning rate metric for policies in both virtual and real scenarios. The tested policy was trained via MAPPO, selecting the policy at the point of 100k episodes of training (training curves for various algorithms are provided in Appendix Figure 8).
> In the virtual scenarios, we conducted 512 episodes of testing, achieving a winning rate of 94.1%.
> As for the real scenarios, we conducted 15 sets of experiments (15 real episodes), achieving a winning rate of 93.3%. The similarity in winning rates between virtual and real scenarios further illustrates UMAP's capability to construct effective virtual training environments as well as to deploy virtual algorithms into real-world scenarios.
>
> We appreciate your suggestions. We will revise the explanation of experimental results in Section 6.4 to avoid potential misunderstandings. And this discussion will also be included in the appendix.
>
> ---
>
> **Q3: Do the authors have any plans to release offline datasets based on the UMAP environment and evaluate the performance of offline algorithms?**
>
> **A3:**
> Yes. As mentioned in the paper, both UMAP and HMAP support all algorithms from the HARL framework, which includes offline algorithms. However, the algorithms in this framework still require interaction between the trained policy and the environment during data collection, meaning they are not entirely offline. Currently, we are testing offline MARL on small-scale tasks and plan to release an offline dataset construction pipeline in the future work.
>
> ---
>
> **Q4：What would be the workload involved in designing a new environment? which parts must be modified through the Unreal Engine, and which can be directly configured through the Python interface?**
>
> **A4:**
> This is a very meaningful question. We have made great effort to ensure the Python interface layer's universality, allowing users to cover modifications of all elements within a POMG directly through this layer.
>
> However, as your example indicates, incorporating customized maps and objects requires modifications to the UE side (the advanced module layer of UMAP). Specifically, rendering materials for maps and objects (such as 3D textures, particle effects) can be obtained from open-source projects in the Unreal Store. The workload depends on the specific modifications but is generally convenient (for example, it took us less than half a day to develop the map and object rendering materials for the *tower-challenge* scenario).
>
> New maps, new objects, new agents, and new properties set can be implemented by inheriting the parent actor class built into UMAP. Thanks to UE's graphical programming language Blueprint, adding new entity types and increasing internal properties can be achieved with minimal effort. After adding the property collection, there is no need to make any changes to the Python side. The Python interface layer can directly recognize the new entity types and properties.
>
> ---
>
> Hope the above responses have clarified our work and addressed your concerns. We **welcome any further questions you might have** and thank you for your dedication!

---

> > ### Comment · Reviewer_1ovJ · 2024-11-27
> >
> > I appreciate the authors for their detailed responses. I have decided to raise my rating by one point.

---

### Official Review · Reviewer_rUB6 · 2024-11-02

**Soundness:** 1
**Presentation:** 2
**Contribution:** 2
**Rating:** 3
**Confidence:** 5

**Summary:**

The authors develop UMAP, an unreal-engine based simulation framework for multi-agent reinforcement learning. They create customisable framework and create a number of tasks based on battle scenarios for evaluating MARL algorithms.

**Strengths:**

The paper does have some strengths:
- The authors develop a new benchmark for multi-agent RL. This is an area with prominent evaluation problems [1] and hence more work on benchmarking is warranted and helpful.
- The use of unreal engine in a benchmark is nice -- it provides better physics simulation than most commonly used multi-agent RL environments.

**Weaknesses:**

However, I have a number of both technical and ethical considerations that I strongly believe should preclude this paper from publication at this conference. I describe my ethical concerns in a separate section and here focus on the paper's technical flaws.

- The arguments that the authors make about MARL benchmarking in the introduction are poor. They claim that MARL benchmarks lack "complexity" and "authenticity" but they don't ever define these terms. Throughout they remain vague reasons to discredit prior work. For example, they claim that SMAC is "authentic" while MPE is not, but recent reimplementations such as SMAX [2] effectively reduce SMAC to a particle environment. Why is MPE not authentic while SMAC is? Is SMAX less authentic than SMAC? Similarly they criticise a lack of complex decision making, but at no point do they demonstrate that their benchmark requires, or even define what this means.
- The introduction also argues that SARL methods performing better than MARL methods implies that algorithms are specialised to environments and struggle to transfer to the real world. The second half of this may be true, but the first does not support it at all! A more realistic interpretation of the first part is that independence is a very strong baseline which multi-agent RL is yet to meaningfully overcome in these types of tasks.
- In line 58 the authors claim that data generated from current benchmarks cannot capture the complexity of real multi-agent scenarios. This is extremely vague and the authors do not back this statement up with any further evidence.
- Their table comparing different MARL environments is also missing some related work, particularly more recent frameworks which have been implemented in JAX [2,3]. JAXMARL, for example, as a framework meets almost all the requirements listed in their table.
- For the flag capture MAPPO environment, they do not seem to clearly explain how the MAPPO agents were trained, or whether they are co-training with the algorithm being evaluated.
- The results in Figure 4 do not follow best evaluation practice [1, 4], which makes the resulting analysis very difficult to believe. For example, they claim that different algorithms do best in different scenarios, but they do not evaluate the aggregate performance! Their likely is an overall best algorithm if you examine, e.g. inter-quartile means calculated with bootstrapped confidence intervals.

**Questions:**

See weaknesses

[1] Gorsane, Rihab, et al. "Towards a standardised performance evaluation protocol for cooperative marl." Advances in Neural Information Processing Systems 35 (2022): 5510-5521.

[2] Rutherford, Alexander, et al. "Jaxmarl: Multi-agent rl environments in jax." arXiv preprint arXiv:2311.10090 (2023).

[3] de Kock, Ruan, et al. "Mava: a research library for distributed multi-agent reinforcement learning in JAX." arXiv preprint arXiv:2107.01460 (2021).

[4] Agarwal, Rishabh, et al. "Deep reinforcement learning at the edge of the statistical precipice." Advances in neural information processing systems 34 (2021): 29304-29320.

**Details Of Ethics Concerns:**

The authors' main contribution is a set of realistic simulated scenarios that aim to replicate battles between autonomous units. The authors then evaluate the ability of algorithms trained in this simulator to transfer to the real world. This clearly and obviously has military applications, which may be the main motivation for the development of these scenarios.

I'd like to note that I have excluded my ethical concerns when scoring this paper, although they are the most compelling reason to reject it from this conference.

I do not believe that papers with such clear and obvious military applications are suitable for publication at this conference.

---

> ### Author Response · Authors · 2024-11-22
> **Author Response to Reviewer rUB6 (Part 1)**
>
> Thank you for your valuable feedback. Here is a detailed, **point-by-point response** addressing your main concerns, minor issues and questions.
>
> ---
>
> **Q1：（Concern about introduction）The authors claim that MARL benchmarks lack "complexity" and "authenticity" but they don't ever define these terms. Throughout they remain vague reasons to discredit prior work.**
>
> **A1：**
> Firstly, we need to clarify that we do not aim to "discredit" the prior works. We fully acknowledge that the rapid development of MARL could not have been achieved without a variety of rich simulation environments.
>
> Our perspective on the "authenticity" of an environment is related to whether a physical engine is used or not, for which the details of a physical engine can be referred to Appendix B. From this standpoint, we mentioned that SMAC "can simulate scenarios of video games" and possesses "a certain degree of authenticity." However, from the perspective of simulating scenarios related to video games, SMAX indeed lacks authenticity, as it only modeling agents as particles in a two-dimensional space and overlooks certain functionalities of game units (such as the Zerg's healing and Protoss's shields).
>
> From the complexity perspective, there are already works indicating that the complexity of existing MARL environments is hard to compare with the complexity of real-world scenarios [1,2]. We use complexity to emphasize the substantial gap between current virtual environments and real-world needs. Our focus is on developing a physical-engine-based environment with strong extendibility, rather than on the built-in tasks. As stated in the paper, "it is unrealistic to develop an all-inclusive environment, but developing an environment that is both extensible and authentic holds significant value." We will make it clear in the introduction to avoid potential misunderstandings regarding the above discussion.
>
> [1] Ming Zhang, Shenghan Zhang, Zhenjie Yang, Lekai Chen, Jinliang Zheng, Chao Yang, Chuming Li,
> Hang Zhou, Yazhe Niu, and Yu Liu. Gobigger: A scalable platform for cooperative-competitive
> multi-agent interactive simulation. In The Eleventh International Conference on Learning Repre-
> sentations, 2022.
>
> [2] Pu Z, Pan Y, Wang S, et al. Orientation and decision-making for soccer based on sports analytics and AI: A systematic review[J]. IEEE/CAA Journal of Automatica Sinica, 2024, 11(1): 37-57.
>
> ---
>
> **Q2: （Concern about introduction）The introduction also argues that SARL methods performing better than MARL methods implies that algorithms are specialised to environments and struggle to transfer to the real world. The second half of this may be true, but the first does not support it at all! A more realistic interpretation of the first part is that independence is a very strong baseline which multi-agent RL is yet to meaningfully overcome in these types of tasks.**
>
> **A2:**
> We partially agree with the reviewer's statement. Since the reviewer mentioned, "independence is a very strong baseline in these types of tasks," and yet these tasks originate from many established MARL benchmarks, doesn't this indicate that some benchmarks are insufficient to reflect the unique challenges of MARL and MAS? In designing the built-in scenarios, our work also focuses on the unique challenges of MARL compared to RL, including the scale perspective, heterogeneity perspective, multi-agent credit assignment perspective, and multi-team perspective. However, the concerns about the introduction mentioned by the reviewer are not significant enough to be considered a weakness of our paper.
>
> After all, we explain in the original text that 1. many practical applications employ classic MARL algorithms or some extended versions of SARL algorithms, and 2. in some MARL test scenarios, MARL algorithms do not outperform SARL algorithms, to illustrate that current MARL algorithms are still not sufficiently developed. Please note that we did not claim in the text that "SARL methods performing better than MARL methods implies that algorithms are specialized to environments and struggle to transfer to the real world," which would be a misinterpretation.
>
> ---
>
> **Q3: (Concern about introduction) In line 58, the authors claim that data generated from current benchmarks cannot capture the complexity of real multi-agent scenarios. This is vague, and the authors do not back this statement up with further evidence.**
>
> **A3:**
>  We appreciate the reviewer pointing this out. We will remove the related statements to prevent possible misunderstandings.

---

> ### Author Response · Authors · 2024-11-22
> **Author Response to Reviewer rUB6 (Part 2)**
>
> **Q4：Their table comparing different MARL environments is also missing some related work, particularly more recent frameworks which have been implemented in JAX. JAXMARL, for example, as a framework meets almost all the requirements listed in their table.**
>
> **A4:**
> We have two fundamental criteria for selecting related works for comparison. The first criterion is that the work should be specifically designed for multi-agent reinforcement learning, and the second is that the work has been peer-reviewed and accepted at significant conferences or journals. The reviewer mentioned JAXMARL, which was accepted by NeurIPS 2024 this year, coinciding with the submission deadline for this conference. We will include JAXMARL in our discussion of related work. However, it is important to note that the reviewer's statement that "JAXMARL as a framework meets almost all the requirements listed in their table" is not accurate. Although JAXMARL incorporates many current MARL benchmarks, it explicitly lacks certain features:
>
> 1. Support for multi-team scenarios. To the best of our knowledge, based on UMAP and HMAP, the *flag-capture* scenario is the first to support multi-team, multi-algorithm training.
> 2. Controllable time flow. Environments in JAXMARL cannot precisely control training and rendering time flows.
>
> Regarding other advantages mentioned in the table, not all benchmarks integrated into JAXMARL possess them. For example, most environments in JAXMARL do not have a physics engine, with only a few environments, such as Multi-Agent Brax, featuring a physics engine, and their engine customization depends on external libraries.
>
> ---
>
> **Q5：For the flag capture MAPPO environment, they do not seem to clearly explain how the MAPPO agents were trained, or whether they are co-training with the algorithm being evaluated.**
>
> **A5:**
> In the *flag-capture* scenario related to *single_mappo* and *double_mappo*, both MAPPO and other algorithms are trained from scratch and tested at fixed intervals. This precisely demonstrates the unique advantage of HMAP, which can evolve multiple MARL algorithms in the same scenario, either from scratch or by loading from a specific model. We appreciate the reviewer's suggestion and will add detailed information to the appendix.
>
> ---
>
> **Q6：The results in Figure 4 do not follow best evaluation practice, the authors do not evaluate the aggregate performance. Their likely is an overall best algorithm if you examine, e.g. inter-quartile means calculated with bootstrapped confidence intervals.**
>
> **A6:** 1. We believe that in the field of RL, the thorough use of random seeds and confidence interval tests is crucial. Based on this notion, we have explicitly employed a 95% confidence interval in our paper, and for this purpose, we have used more than 10 random seeds in some tasks.
>
> 2. By analyzing the current experimental data, it is possible to identify an algorithm that achieves the best aggregate performance across all tasks. However, this does not mean that the algorithm is the best in every single task. Our core argument is that the existing SOTA algorithms cannot perform best in all tasks, which implies the need for more advanced algorithms. **Evaluating the aggregate performance and obtaining a current overall best algorithm does not significantly aid in proving our argument.**

---

> ### Author Response · Authors · 2024-11-22
> **Author Response to Reviewer rUB6 (Part 3)**
>
> **Q7: Ethics Concerns**
>
> **A7:**
> The reviewer mentioned our work involves content related to battle and implied that our main motivation originates from military application needs. **This is a very serious accusation.** Honestly, we are shocked by such ethical concerns and believe **it is necessary to clarify immediately**. We will address this issue from three perspectives: the origins of our task designs, the virtual assets used, and the physical components involved.
>
> 1. All task designs in UMAP are inspired by various games.
>
> Specifically, *metal-clash*, our first scenario created using UMAP, draws inspiration from *StarCraft* and *SMAC*.
>
> *Flag-capture* is inspired by the 3D multiplayer first-person video game *Quake III Arena* and its AI research [1].
>
> *Tower-challenge* is inspired by the reverse tower defense game *Anomaly: Warzone Earth*, where players act as the attackers.
>
> *Landmark-conquer* is inspired by research on agent roles [2], which modifying the SMAC scenario where allied troops need to protect an allied building from enemy troops.
>
> 2. All virtual materials used in UMAP directly come from open-source projects in the Unreal Store (now renamed Fab). We have focused our main efforts on architecture development, adaptation with MARL, and optimizing user experience, rather than designing textures, particle effects (term in UE), and other materials.
>
> 3. All physical materials mentioned in the paper, such as robots, drones, and obstacles, come from the famous international robot competition ICRA RoboMaster, which are unrelated to any military applications.
>
> In summary, the built-in tasks developed based on UMAP are all inspired by computer games, and both virtual and physical materials come from open-source civilian community domains. Claiming that we have “clearly and obviously military applications” is inappropriate. However, we appreciate your kind reminder and will add a checklist for ethical review as proposed by Thorny Roses [3] in the appendix of our paper.
>
> [1] Jaderberg M, Czarnecki W M, Dunning I, et al. Human-level performance in 3D multiplayer games with population-based reinforcement learning[J]. Science, 2019, 364(6443): 859-865.
>
> [2] Nguyen D, Nguyen P, Venkatesh S, et al. Learning to Transfer Role Assignment Across Team Sizes. Proc. of the 21st International Conference on Autonomous Agents and Multiagent Systems (AAMAS 2022)
>
> [3] Kaffee L A, Arora A, Talat Z, et al. Thorny Roses: Investigating the Dual Use Dilemma in Natural Language Processing[J]. arXiv preprint arXiv:2304.08315, 2023.

---

> > ### Comment · Reviewer_rUB6 · 2024-11-25
> > **Thank you for your response**
> >
> > I'd like to thank the authors for their response to my questions and concerns.
> >
> > I will start by addressing my ethics concerns around this paper. I acknowledge the seriousness of raising this paper for ethics review, and would not do so frivolously. The main argument the authors make is that the materials are all open-source. While I understand this, I do not agree that this means their work does not have military applications. My claim is that building a simulator for battle scenarios and then demonstrating that policies trained in this simulator can transfer to the real world is unethical, even if the robots themselves cannot cause harm, because of the clear potential for military applications based on this. I think that ICLR, a machine learning conference, is not the right venue for such work. The competition linked by the authors is also explicitly a competition where robots battle against one another in a combat scenario. However popular this competition, it also clearly is aimed at increasing military capabilities of robots.
> >
> > I am not at all persuaded by the authors' claims about their works, but I also acknowledge that I am not an ethicist or heavily involved in AI ethics. I will therefore maintain my decision to raise this paper for ethics review, where experienced ethics reviewers can have the final say about this aspect of the paper.
> >
> > I will now address some of the other points that the authors made.
> >
> > Q1: I think if anything, the justification the authors make here underlines my point. While e.g. SMAX does not have some aspects of StarCraft, it seems odd to claim that there is some *fundamental* difference between these two environments because one of them has e.g. health regeneration and the other doesn't. Complexity, even after the clarification of the authors, seems similarly vague. I thank the authors for their promise to address these issues in a new version of the introduction.
> >
> > Q2: I disagree with the authors here. Specifically they state
> >
> > > doesn't this indicate that some benchmarks are insufficient to reflect the unique challenges of MARL and MAS?
> >
> > I agree that evidence suggests prior work does require specialist multi-agent solutions. However, I am not convinced that the authors' new benchmark addresses those concerns. What is so different from this work to the numerous other game-based multi-agent RL environments that means that it cannot be solved by e.g. MAPPO and a reasonable number of samples? Indeed, looking at the authors' results, success rates are around 80% for almost all of the scenarios they design.
> > I think there are a lot of open questions here that are being swept under the rug. Whether you *actually* need multi-agent methods is, in my opinion, a very much open question! I therefore don't really accept the authors' reasoning here, but also acknowledge that there are more important issues with the paper.
> >
> > Q3: Thanks for addressing this.
> >
> > Q4: Thank you for clarifying your criteria for inclusion in the table, which are fine by me. I agree that JaxMARL does not meet all the requirements (hence the use of almost!) and with the authors' points where it does not support all the features their work does.
> >
> > Q5: Thank you for adding clarification.
> >
> > Q6: I don't agree with the authors here. While I appreciate the authors' use of 10 seeds and 95% confidence intervals, the standard way of aggregating performance in Deep RL is via bootstrapped confidence intervals and the inter-quartile mean. As the authors are proposing a benchmark paper, their paper will be used as the standard for how to evaluate on this set of tasks in the future. As such, a lack of aggregation among the algorithms isn't acceptable. Secondly, the authors claim that there is significant room for new algorithms and that current approaches are sub-optimal. However, in almost all scenarios an algorithm gets a score of above 80%. In fact only in tower_challenge_hard do the algorithms make little progress. This suggests that the current challenges suggested by the authors do not have that much headroom.

---

> > > ### Author Response · Authors · 2024-11-25
> > > **Author Response to Reviewer rUB6 (New Part)**
> > >
> > > Thank you for your response. We are delighted to have addressed some of your concerns (even if only partially).
> > >
> > > ---
> > >
> > > 1. We are once again shocked by the reviewer's comments regarding the ethics review. **Our work is a general-purpose environment that allows users to develop a variety of tasks, with the built-in environments being part of our contribution but not the main focus**. Additionally, we have listed the inspiration behind each scenario's design in our response, all of which are derived from video games, and not all scenarios are battle-based (like tasks in SMAC). Lastly, the reviewer's claim that the ICRA Robomaster, an international robotics competition, “clearly is aimed at increasing military capabilities of robots” is confusing. *This seems to be a bias against the organizer ICRA and the competition itself.* We refrain from further comment on this matter.
> > > 2. Regarding the scalability based on UMAP, the tasks generated from built-in scenarios are not fixed, and their corresponding difficulty levels can be adjusted. We have repeatedly pointed out in the paper that each scenario's task is fully customizable, naturally allowing for difficulty adjustments through modifications of the POMG elements. Just as the difference between the SMAC's *MMM* map and the *MMM2* map (where task difficulty is increased by adding enemy units), **all scenarios in UMAP can adjust task difficulty through modifications of agent numbers, individual agent properties (observation range, movement speed, HP, etc.), and initial agent positions, which is quite evident.** Therefore, the reviewer's claim that the “scenarios” are not particularly challenging is inappropriate.
> > >
> > > ---
> > >
> > > Thank you for all the efforts you have put into reviewing our paper.

---

### Official Review · Reviewer_cWdv · 2024-11-03

**Soundness:** 4
**Presentation:** 4
**Contribution:** 3
**Rating:** 8
**Confidence:** 4

**Summary:**

Overall Summary:
- Proposes and provides open source implementation of
	- UMAP - new MARL environment that combines features of multiple other environments into one (Table 1)
	- HMAP - experimental framework that has implementations of popular MARL algorithms (to be paired with UMAP)
	- Experimental results on the application of 7 popular MARL algorithms on UMAP environments
	- Sim to real interaction between UMAP and real world hardware and scenarios.


UMAP Summary:
- Unreal engine base, but users interact with higher layers (state, action, observation etc.)
- Tasks and scenarios:
	- Metal clash - cooperative/competitive - two teams battle in a team elimination type game
	- Tower challenge - full cooperative - agents work together to destroy a central objective
	- Flag capture - cooperative/competitive - multiple teams aim to capture a single flag
	- Landmark conquer - cooperative/competitive - attack and defend with two teams.
- All scenarios have fully editable parameters including but no limited to: team size, number of teams, health and other attributes on each player within a team

HMAP Summary:
- An entire experimental framework.
- contains implementations of popular MARL algorithms
- Integrates UMAP as well as other popular environments (StarCraft, multi-particle, gym etc) with popular MARL algorithms by providing training scripts
	- Authors provide a description of their "glue module" system for training.
	- Looking at the appendix D, the glue module appears to make training extremely user friendly by allowing easy assignment of different algorithms to different tasks and teams.

Experimental Results summary:
- Interpreted this section as a proof of concept for UMAP and HMAP
- Every task has its own best algorithms
- Biggest takeaway is that the UMAP scenarios are learnable by popular MARL algorithms
- Second biggest takeaway is the demonstration that HMAP experimental framework works well in training popular algorithms on UMAP

Sim to real summary:
- Proof of concept for bridging simulation training to real time execution.
- Agents trained on UMAP and then controlled objects in real-world scenarios.
- Created a system by which the physical hardware communicates with UMAP to update UMAP's internal environment, and then UMAP and HMAP communicate to form a decision. The decision is then passed back to the physical hardware to execute an action.

**Strengths:**

Originality:
- Roots its contributions very well in previous works. Provides the reader with a sense of improvement and expansion of the field.
- Takes positive benefits from contemporary environments and combines it into a single environment.
- Introduces time-flow speed that should aid in the speed up of training MARL algorithms, as well as the debugging process.

Quality:
- Overall high quality writing.
- Figures and tables clearly support the claims made in the paper.
- Committed to open-source work and development after review process

Clarity:
- Ideas conveyed clearly and concisely
- Hyperparameters provided in a clear manner
- Diagrams and figures detailing UMAP, results on UMAP, and glue module are intuitive

Significance:
- Provides both the UMAP environments and scenarios as well as a training framework for them, HMAP
- HMAP framework also works for previous environments as well.
- Time in UMAP (Appendix C) and time flow control. Separates real time flow and simulation time flow and gives control to user for different purposes. In training, user will likely speed up simulation time to address a major issue in MARL, the need for a lot of data collection
- User can also slow down simulation time for debugging. Very useful feature overall

**Weaknesses:**

Weaknesses:
- UMAP and HMAP did not appear to support agent to agent communication. This notion is available in previous environments such as MPE. As teams get larger, it becomes less feasible to use a central controller for all players on a team and thus communication between individual agents would become more important

- the diversity of tasks is quite limited in the current pool of UMAP scenarios. Would enjoy seeing a more diverse set of environments outside of largely combat based team tasks. Examples of this include but are not limited to warehouse management scenarios, hide and seek, and other non-combat scenarios primarily those missing physics simulation environments

- Training time will differ from hard ware to hardware, but it would be good to see how long K episodes of training in a UMAP scenario takes versus K episodes of training in a StarCraft environment (assumes episode lengths are equal)

	- Providing such a table looking at run times with the same hardware in an appendix section can give potential users a sense of the speed up or slow down UMAP provides.

	- Along a similar vein, it would be nice to have a table or figure exploring how much real time is saved by utilizing the fast time dilation factor within UMAP scenarios.

	- All in all, I recognize that the paper has made strides towards speeding up training appropriately for different hardware, but I have little idea on the expected training time compared to that of other environments from the paper itself.

**Questions:**

My questions are directly related to certain points from the Weaknesses section:
- Given any fixed hardware, how slow or fast does it take to train an agent in UMAP as it does another physics-engine environment, such as SMAC?

- Are there plans to include more diverse types of scenarios in UMAP outside of health based combat scenarios?

- How much time does the time-dilation feature save in training a fixed agent, such as MAPPO?

---

> ### Author Response · Authors · 2024-11-21
> **Author Response to Reviewer cWdv (Part 1)**
>
> Thank you for your valuable feedback. Here is a detailed, **point-by-point response** addressing your main concerns, minor issues and questions.
>
> ---
>
> **Q1: Do UMAP and HMAP support agent communication?**
>
> **A1:**
> Although our paper does not explicitly mention scenarios involving multi-agent communication, both UMAP and HMAP do support it. From the perspective of MARL modeling, incorporating a communication mechanism essentially alters the agents' "general" observation function (to compensate for the original limitations of partial observability) and the form of the policy (by adding communication modules, or changing centralized policies to distributed, etc.), all of which can be customized in UMAP and HMAP. Since the UMAP's UE side and Python side transmit the entire global information of the environment, converting a built-in task into one similar to *Cooperative Communication* task in MPE does not require modifications on the UE side. It only necessitates adjustments at the fifth layer of the UMAP Python interface and on the algorithmic side. The former modifies the mapping from the global state to the information input into each agent's policy network, while the latter changes the structure of the agent's policy.
>
> ---
>
> **Q2：The diversity of tasks is quite limited in the current pool of UMAP scenarios. Are there plans to include more diverse types of scenarios in UMAP outside of health-based combat scenarios?**
>
> **A2:**
> Here, we hope to address your second concern and your second question together. In developing UMAP's built-in scenarios, our focus was more on identifying the unique challenges that MARL presents compared to RL, rather than the *themes* of the scenarios themselves. Specifically, these challenges arise due to transitions:
> - from a single agent to multiple agents (scale perspective),
> - from homogeneous agent to heterogeneous agents (heterogeneity perspective),
> - from individual reward to group rewards (credit assignment perspective),
> - from a single objective to mixed objectives (multi-team perspective).
>
> Therefore, we believe the diversity of our tasks is reflected in the academic challenges of MARL, rather than the specific themes of the tasks. Thank you for your suggestion! We will continue to maintain the UMAP repository in our future work and add more built-in tasks with various themes.

---

> ### Author Response · Authors · 2024-11-21
> **Author Response to Reviewer cWdv (Part 2)**
>
> **Q3: Compared to the StarCraft environment, how long does it take to train the same number of K episodes in UMAP? Given the same hardware configuration, how does the training duration in UMAP compare to that in another physics-engine environment like SMAC? Additionally, how much time does the time-dilation feature save when training a fixed agent?**
>
> **A3:**
> This is a meaningful question, and we will answer it together with your first and third questions. To compare the actual time consumed by UMAP and SMAC in training the same episodes, we utilized the HMAP framework to run the same algorithms in similar tasks of UMAP and SMAC, observing the differences in training time required. We also compiled statistics on resource consumption indices and efficiency index under different time dilation factors.
>
> To ensure a fair comparison, we set up the experiments as follows:
>
> -  Both UMAP and SMAC had same parallel process counts set to 8, with the maximum length of each episode set to 150 timesteps.
> - UMAP employed the *metal_clash_5sd_5mc* task, while SMAC used the *MMM* task. These are two similar tasks, each involving 20 heterogeneous agents.
> - To eliminate the influence of algorithmic differences, we conducted experiments on the QMIX algorithm from PyMARL2 and the MAPPO from HARL, maintaining consistent hyperparameters across both.
> - All experiments were conducted on a Linux server equipped with AMD 7742 CPU (maximum frequency 2.25GHz) and NVIDIA RTX3090 GPUs. At the beginning of each experiment, the server was maintained in an idle state, executing only the essential system processes.
>
> It is worth mentioning that even though the maximum lengths of all episodes are the same, some episodes may end prematurely, leading to significant variability in the lengths of different episodes. **To avoid this, we did not calculate how long it took to train every K episodes; instead, we recorded the time taken for the same number of timesteps, calculating FPS (virtual timestep per real second) as the efficiency index.**
>
> Below are the results of running the MAPPO algorithm with varying time dilation coefficients under the same number of parallel environments:
>
> **HARL-MAPPO-8-parallel-envs**
>
> | **Time Dilation Factor** | **0.5** | **1.0** | **2.0** | **4.0** | **8.0** | **16.0** | **32.0** | **64.0** | *SMAC* |
> |--------------------------|---------|---------|---------|---------|---------|----------|----------|----------|----------|
> | FPS(virtual timestep per real second) | 7.46 | 14.43 | 29.08 | 56.89 | 123.80 | 191.53 | 257.35 | 435.66 | *144.57* |
> | CPU Utility(%) | 0.1 | 0.2 | 0.4 | 0.7 | 1.1 | 1.4 | 1.6 | 1.9 | *1.1* |
> | Memory (MB) | 4826 | 4804 | 4854 | 4895 | 4906 | 4969 | 4978 | 4944 | *8371* |
> | GPU Memory (MB) | 1639 | 1639 | 1639 | 1639 | 1639 | 1639 | 1639 | 1639 | *1680* |
>
>
> Below are the results of running the QMIX algorithm with varying time dilation coefficients under the same number of parallel environments:
>
> **PyMARL2-QMIX-8-parallel-envs**
>
> | **Time Dilation Factor** | **0.5** | **1.0** | **2.0** | **4.0** | **8.0** | **16.0** | **32.0** | **64.0** | *SMAC* |
> |--------------------------|---------|---------|---------|---------|---------|----------|----------|----------|----------|
> | FPS(virtual timestep per real second) | 10.17 | 22.78 | 40.82 | 82.33 | 158.76 | 285.31 | 384.19 | 643.43 | *247.02* |
> | CPU Utility(%) | 0.1 | 0.2 | 0.4 | 0.6 | 1.1 | 1.6 | 1.7 | 1.9 | *1.2* |
> | Memory (MB) | 4826 | 5034 | 5158 | 5176 | 5225 | 5312 | 5358 | 5585 | *11103* |
> | GPU Memory (MB) | 1531 | 1531 | 1531 | 2061 | 2139 | 2161 | 2161 | 2161 | *5828* |
>
> Based on the results above, we can draw the following conclusions:
>
> 1. With a time dilation coefficient >=16, UMAP's training efficiency on similar tasks surpasses that of SMAC.
> 2. As the time dilation coefficient increases, both FPS and CPU utilization see a significant rise, while memory and GPU memory usage remain relatively unchanged.
> 3. Within a certain range, FPS and CPU utilization are directly proportional to the time dilation coefficient. When the coefficient reaches a certain threshold (determined by the maximum frequency of a single-core CPU), this relationship degrades into a positive correlation.
>
> ---
>
> Hope the above responses have clarified our work and addressed your concerns. We **welcome any further questions you might have** and thank you for your dedication!

---

### Official Review · Reviewer_1Wxb · 2024-11-04

**Soundness:** 2
**Presentation:** 3
**Contribution:** 3
**Rating:** 6
**Confidence:** 5

**Summary:**

The paper introduce Unreal Multi-Agent Playground (UMAP), a 3D simulation environment for multi-agent reinforcement learning, built on Unreal Engine. UMAP addresses existing limitations by offering a flexible platform for complex tasks and research involving diverse agents and teams. Alongside UMAP, they present the Hybrid Multi-Agent Playground (HMAP), compatible with various algorithms. The paper showcases UMAP's capabilities through tasks, evaluations of MARL algorithms, and a sim-to-real experiment.

**Strengths:**

1. Addressing a Significant Gap: The paper identifies and addresses a critical limitation in MARL research—the lack of simulation environments that are both realistic and capable of modeling complex, large-scale multi-agent interactions. By providing a physics-based environment, UMAP brings simulations closer to real-world scenarios, which is essential for the development and evaluation of practical MARL algorithms.
2. High Extensibility and Customization: UMAP's hierarchical, modular architecture allows users to easily customize tasks at various levels, from high-level configurations to low-level implementations. This flexibility enables researchers to create a diverse range of scenarios tailored to specific research questions or application domains.
3.Support for Diverse and Complex Tasks: The inclusion of scenarios featuring heterogeneous agents, large-scale populations, multiple teams, and sparse rewards demonstrates UMAP's ability to model a wide spectrum of complex MARL problems. This diversity is valuable for testing the robustness and generalization capabilities of MARL algorithms.
4.Open-Source Contribution: By releasing UMAP and HMAP as open-source projects, the authors contribute valuable tools to the MARL community. This openness encourages collaboration, reproducibility, and further development by other researchers.

**Weaknesses:**

1. Depth of Experimental Analysis: The experimental results primarily focus on win rates and rewards. A deeper analysis of algorithm behaviors would provide more comprehensive insights into the challenges posed by the new environment.
2. Comparison with Existing Environments: Although the paper mentions limitations of current MARL environments, a more detailed comparison highlighting specific features and performance metrics would better contextualize UMAP's advantages. Including benchmarks or case studies demonstrating UMAP's superiority could strengthen the argument.
3. Accessibility and Cost of Use: While UMAP is described as user-friendly, potential barriers such as the need for familiarity with Unreal Engine and the complexity of setting up the environment could limit adoption. Providing comprehensive tutorials, documentation, and support could mitigate this issue. In addition, Physics-based environments, especially those built on game engines like Unreal Engine, can have significant computational overheads. Discussion about the performance implications, resource requirements, and optimizations would be beneficial for users with limited computing resources.

**Questions:**

1. Though it is a general platform, what's the special strength of the new environment built on this platform for evaluating existing MARL algorithms?
2. How much effort should a new developer paid for introducing his own environment/problem into this platform?

---

> ### Author Response · Authors · 2024-11-21
> **Author Response to Reviewer 1Wxb (Part 1)**
>
> Thank you for your valuable feedback. Here is a detailed, **point-by-point response** addressing your main concerns, minor issues and questions.
>
> ---
>
> **Q1: The experimental section primarily focuses on win rates and rewards. A deeper analysis of algorithm behaviors would provide more comprehensive insights into the challenges posed by the new environment.**
>
> **A1:**
> We believe the most valuable contribution of UMAP lies not in the built-in environments developed based on it, but in its substantial extendibility and its capability to simulate real-world physical characteristics. In fact, by designing this series of learnable built-in environments, we demonstrate UMAP's potential to create MARL tasks with unique challenges (such as heterogeneity, large scale, and sparse team rewards) that are distinct from those in SARL. On these built-in tasks, we believe that a detailed analysis of policy behavior may not be necessary. **Win rates and rewards are sufficient to prove our core argument: no current SOTA algorithm can achieve best performance across all tasks. Therefore, there is a need to design new tasks and test new algorithms in highly extensible
> environments.**
>
> We appreciate your suggestion and consider developing more built-in scenarios with scenario-specific metrics in our future work.
>
> ---
>
> **Q2: While the paper mentions the limitations of current MARL environments, a more detailed comparison highlighting specific features and performance metrics would better contextualize UMAP's advantages.**
>
> **A2:**
> Thank you for your suggestion! Unlike papers focused on algorithms, it's inherently challenging for papers focusing on environments to compare using performance metrics, especially when the environment possesses many unique functionalities. However, the suggestion to incorporate case studies to illustrate UMAP's advantages is valuable.
>
> As mentioned in the paper, beyond supporting a diversity of tasks (see *Q4&A4* for design principle details), UMAP's advantages also include its *fully customizable design based on a physics engine*, *controllable time flow*, and *rich rendering mechanisms*.
> 1. We contextualize the advantages related to customization by introducing UMAP's hierarchical design and its relationship with the customization of various POMG elements.
> 2. In fact, regarding the benefits of controllable time flow, we have conducted experiments on the relationship between time dilation factors and resource consumption/training efficiency, and also recorded tutorial videos for slow-motion analysis by controlling the time dilation factor. We will add these experiments to the appendix.
> 3. Concerning UMAP's rich rendering mechanisms, we already have ample videos showcasing a) rendering in the UE editor, b) on a compiled pure rendering client, c) on a compiled training client, and d) cross-platform real-time rendering. However, these are not easily reflected in the paper. We consider adding related explanatory videos and web links to the appendix after the review process.

---

> ### Author Response · Authors · 2024-11-21
> **Author Response to Reviewer 1Wxb (Part 2)**
>
> **Q3：For users unfamiliar with Unreal Engine, UMAP presents potential usage barriers. Offering tutorials and support would help mitigate this issue. On the other hand, environments developed with UMAP might incur significant computational costs, necessitating consideration for users with limited computing resources.**
>
> **A3:**
> We agree on the importance of *Accessibility and Cost of Use* for the widespread adoption of an MARL environment.
>
> **Concerning the potential barriers to customizing environments**, we have implemented a hierarchical design in our environment architecture. Users can focus solely on the Python layer for full customization of the task elements. Additionally, for users who need to customize at the UE layer, such as building a digital twin system for their specific task scenarios, we offer comprehensive tutorials. In fact, in our open-source program, we have already prepared detailed tutorials for UMAP and HMAP, including tutorial PPTs and demonstration videos.
>
> **Regarding concerns about the computational overheads of UMAP**, we also made significant efforts to make UMAP accessible to users with various computational resources.
> - In terms of compatibility with computing platforms and operating systems, UMAP supports training on Linux, Windows, and MacOS, and accommodates training on computing devices without GPUs.
> - Moreover, in some cases, users' GPU, CPU, and memory resources may be unbalanced, leading to waste when one resource reaches its limit while others remain underutilized. UMAP addresses this by allowing adjustments to the time dilation factor and the number of parallel processes to fully utilize the user's computing resources. The number of processes can be adjusted to manage memory and GPU memory resources, while the time dilation factor can be used to maximize CPU resource utilization without changing memory-related resources, thereby increasing the training speed of the environment.
> - We have also conducted experiments on resource consumption to demonstrate the computational efficiency optimizations of UMAP. The entire experiment was conducted on a Linux server equipped with AMD 7742 CPU（2.25GHz）and NVIDIA RTX3090 GPUs. In the *metal_clash_5sd_5mc* task, with a time dilation factor of 32 and a parallel envs count of 8, training the QMIX algorithm from PyMARL2 only occupied 1.7% of the CPU, 5.36GB of memory, and 2.16GB of GPU memory, and required only 80s to train 1024 episodes. This implies that **a server equipped with consumer-grade graphics cards can support up to 50 such training tasks simultaneously, with each task requiring less than 3 hours to train to 100k episodes**. For more detailed information, please refer to our response to Reviewer cWdv（Part 2）.
> We appreciate your suggestions and will include this discussion in the appendix of our paper.

---

> ### Author Response · Authors · 2024-11-21
> **Author Response to Reviewer 1Wxb (Part 3)**
>
> **Q4： Though it is a general platform, what's the special strength of the new environment built on this platform for evaluating existing MARL algorithms?**
>
> **A4：**
> UMAP is an extensible environment specifically designed for MARL. The design philosophy of its built-in environments is to reflect the unique challenges of MARL compared to single-agent RL. These challenges arise due to transitions：
> - from single agent to multiple agents (*scale perspective*),
> - from homogeneous agents to heterogeneous agents (*heterogeneity perspective*),
> - from individual reward to group rewards (*credit assignment perspective*),
> - from single objective to mixed objectives (*multi-team perspective*).
>
> These built-in environments are designed to specifically highlight each challenge. For example, in a series of tasks designed around the *metal-clash* scenario focusing on heterogeneity, each type of agent has completely different observation spaces and functions, characterized by: 1. Different observation ranges, 2. Different numbers of entities that can be recorded, 3. Different precision in recording each entity. This is in contrast to classic environments like SMAC, where heterogeneous agents share similar observation spaces and functions. Moreover, each type of agent has different action spaces (number of discrete actions), and the properties of agents can be further modified and extended to create more types of heterogeneous agents.
>
> Similarly, in *metal-clash* tasks focusing on large-scales, the built-in scripts and environments of agents are fully considered, introducing observation interference and activity restrictions to provide convenience for the emergence of complex group policies. This differs from classic environments like MAgent, where larger groups always have a decisive advantage over smaller ones. In UMAP's tasks, agents can leverage environmental factors to achieve victories with fewer numbers.
>
> In the *tower-challenge* scenario, which focuses on sparse team rewards, agents need to learn to organize formations and attack the defense tower at the right time to receive rewards. This represents a huge exploration space and very sparse rewards, somewhat similar to the challenging tasks in GRF where rewards are only given for scoring goals. However, compared to GRF, the difficulty of *tower-challenge* tasks can be smoothly adjusted by tweaking the parameters of the defense towers and agents.
>
> Regarding multi-team scenarios, to the best of our knowledge, the *flag-capture* scenario of UMAP is the first to support multi-algorithm and multi-team training. Two teams may cooperate in the early stages of an episode to combat other teams, and compete in the later stages for the final victory.
>
> Overall, these tasks developed based on UMAP not only can clearly reflect the aforementioned unique challenges, but also have customizable features to generate a richer variety of tasks.

---

> ### Author Response · Authors · 2024-11-21
> **Author Response to Reviewer 1Wxb (Part 4)**
>
> **Q5：How much effort should a new developer paid for introducing his own environment/problem into this platform?**
>
> **A5:**
> This is a meaningful question. To answer this, let's first clarify the basic workflow for a new developer using UMAP, which includes: *Environment Setup*, *Task Development*, *Algorithm Training*, *Result Rendering*, and *Feedback for Further Development*.
>
> Notably, the effort spent on *Result Rendering* is independent of the specific tasks. Developers can master the full set of UMAP rendering mechanisms by spending less than an hour watching our tutorial videos, as no secondary development on the rendering mechanism is required, only straightforward operations. For *Environment Setup*, *Task Development*, and *Algorithm Training*, we will analyze both the simplest and the most complex cases.
>
> In the simplest case, developers do not need to make any changes on the UE side. For *Environment Setup*, they only need to deploy our pre-configured Docker image, which includes compiled binary files, HMAP code, and a series of environment
> configurations, achievable within minutes. For *Task Development*, utilizing built-in scenarios and agents through the Python interface layer allows for quick customization of tasks. As for *Algorithm Training*, users can choose their familiar third-party code frameworks and directly copy the algorithm code into HMAP's third-party code folder.
>
> In the most complex case, such as when users need to develop entirely new scenarios and conduct physical experiments for validation, the *Environment Setup* step requires installing Unreal Editor and our modified Unreal Engine (UMAP's native and specification layers), which can typically be completed within two hours under tutorial guidance. For *Task Development*, users can design and develop agents and maps using the rich resources available in the Unreal Game Store, usually within a week. If further integration with physical experiments is needed, the development of kinematics and dynamics for agents, action design, and behavior tree design may require more effort. However, thanks to UE's simple and practical Blueprint language, such tasks can typically be completed within a month. *Algorithm Training* in complex cases is similar to the simple case, where users only need to merge the code from their familiar third-party frameworks into HMAP.
>
>
> ---
>
> Hope the above responses have clarified our work and addressed your concerns. We **welcome any further questions you might have** and thank you for your dedication!

---

> > ### Comment · Reviewer_1Wxb · 2024-11-24
> > **Discussion**
> >
> > Thank you for your detailed response and the additional information provided. I believe these updates partially address my concerns, and as a result, I have increased my review score to 6. However, as noted by other reviewers, your work lacks sufficient analysis and comparison with existing MARL benchmarks. While it is not necessary to conduct a detailed comparison with existing works, it is crucial to clearly articulate the key unique aspects of your approach, particularly in the context of MARL.
> >
> > I will provide further detailed comments at a later time to assist the authors in enhancing this work. I hope these suggestions will help improve the overall quality and impact of your research.

---

> > > ### Author Response · Authors · 2024-11-25
> > > **Author Response to Reviewer 1Wxb**
> > >
> > > We sincerely appreciate the reviewer's recognition of our existing work and the suggestions for improvement. We are very much looking forward to your further advice! In the meantime, we will make certain modifications to the prior work mentioned in the related work section of our paper and elaborate more on the unique features of our work. We will ensure that a complete revised PDF is uploaded before the end of the discussion phase. Thanks!

---

### Author Response · Authors · 2024-12-03

Dear Reviewers,

We would like to express our sincere gratitude to all reviewers for your dedicated time, constructive comments, and valuable feedback on our paper. Your suggestions and recommendations have helped us further clarify and strengthen the presentation of our work.

As we approach the end of the discussion phase, we kindly invite you to share any additional questions or concerns you may have. We are committed to addressing all remaining issues and would greatly appreciate your further input in the discussion phase.

We look forward to any additional feedback you may wish to provide.

---

### Author Response · Authors · 2024-12-04
**Overall Rebuttal**

We thank all the reviewers for their valuable suggestions and engagement during the discussion phase. Your time and effort have clarified our work, and we sincerely appreciate the constructive feedback.

We are encouraged by the reviewers' appreciation that **UMAP addresses a significant gap in the MARL domain by incorporating Unreal Engine for improved physical simulation conditions** (1Wxb, rU86, 1ovJ), that **UMAP is highly scalable** (1Wxb, tZCz) and **supports a diverse range of task simulations** (1Wxb, 1ovJ, tZCz), that the collaboration between HMAP and UMAP, as well as **HMAP's compatibility with other frameworks, is very useful** (cWdv, MxdE), that **real-world experiments have shown UMAP's potential in sim-to-real applications** (tZCz), and that **our series of open-source plans will benefit the community** (1Wxb, cWdv). In terms of manuscript writing, we also thank the reviewers for praising the **high quality of our writing** (cWdv, 1ovJ), the clarity of our viewpoints, and providing **detailed experimental settings and clear charts** (cWdv).

To help address the concerns of the reviewers and facilitate further discussion, we have revised the PDF version of our manuscript. The revisions include:
1. Incorporating additional related work on MARL simulation environments, refining the presentation of UMAP's unique advantages, and enriching the details in Table 1.
2. Offering a detailed discussion on various related works in the appendix, clearly identifying features lacking in each work compared to UMAP.
3. Expanding the appendix to include a detailed introduction to UMAP's built-in scenarios.
4. Expanding the appendix to include a discussion on UMAP's training efficiency and resource consumption.
5. Introducing a checklist for ethical evaluation of the paper in the appendix.
6. Refined the expression of certain details within the paper.

---

### Meta-Review · Area_Chair_kuRj · 2024-12-19

**Metareview:**

This paper presents UMAP, a physics-based simulation environment built on Unreal Engine for multi-agent reinforcement learning.
The paper suffers from fundamental positioning ambiguity (unclear benchmark vs platform focus), methodological shortcomings (insufficient comparative evaluation, weak validation of sim-to-real claims), and inadequately substantiated arguments against existing MARL frameworks. The empirical results fail to demonstrate compelling advantages over existing solutions, while concerns persist about accessibility and adoption barriers due to implementation complexity. These critical limitations, highlighted particularly by Reviewers rUB6 and MxdE, suggest the work requires substantial revision to meet acceptance standards.

Given these limitations, I recommend rejection with encouragement to address these issues in a future submission.

**Additional Comments On Reviewer Discussion:**

While most reviewers acknowledge its technical merits in providing extensible scenarios and bridging sim-to-real gaps, ethical concerns have been raised regarding potential military applications, particularly given the battle-based scenarios and sim-to-real capabilities.

---

### Decision · Program_Chairs · 2025-01-22

Reject